# OFFLINE PREFERENCE-BASED VALUE OPTIMIZATION

**Hyungkyu Kang**
Seoul National University, Upstage
Seoul, South Korea
hyungkyu0119@snu.ac.kr

**Min-hwan Oh**
Seoul National University
Seoul, South Korea
minoh@snu.ac.kr

## ABSTRACT

We study the problem of offline preference-based reinforcement learning (PbRL), where the agent learns from pre-collected preference data by comparing trajectory pairs. While prior work has established theoretical foundations for offline PbRL, existing algorithms face significant practical limitations: some rely on computationally intractable optimization procedures, while others suffer from unstable training and high performance variance. To address these challenges, we propose *Preference-based Value Optimization* (PVO), a simple and practical algorithm that achieves both strong empirical performance and theoretical guarantees. PVO directly optimizes the value function consistent with preference feedback by minimizing a novel *value alignment loss*. We prove that PVO attains a rate-optimal sample complexity of $\mathcal{O}(\varepsilon^{-2})$, and further show that the value alignment loss is applicable not only to value-based methods but also to actor–critic algorithms. Empirically, PVO achieves robust and stable performance across diverse continuous control benchmarks. It consistently outperforms strong baselines, including methods without theoretical guarantees, while requiring no additional hyperparameters for preference learning. Moreover, our ablation study demonstrates that substituting the standard TD loss with the value alignment loss substantially improves learning from preference data, confirming its effectiveness for PbRL.

## 1 INTRODUCTION

One of the major challenges in reinforcement learning (RL) is designing suitable reward functions for real-world tasks. Reward design often requires costly instrumentation such as motion capture (Akkaya et al., 2019; Peng et al., 2020), and poorly designed reward functions can significantly degrade training performance. Preference-based reinforcement learning (PbRL) provides a compelling alternative by inferring the underlying reward signal from preference feedback, such as human comparisons between trajectories (Christiano et al., 2017). This framework has demonstrated its effectiveness in domains where direct reward specification is difficult, including robotics (Brown et al., 2019; Shin et al., 2023), games (MacGlashan et al., 2017; Warnell et al., 2018), and language models (Ziegler et al., 2019; Stiennon et al., 2020; Ouyang et al., 2022).

In this work, we study offline PbRL, where learning is conducted using pre-collected datasets of trajectories and preference feedback (Kim et al., 2023; An et al., 2023; Hejna & Sadigh, 2024). Offline RL (Levine et al., 2020) is advantageous in scenarios where real-time online interaction may be costly or unsafe. This consideration is especially relevant for PbRL, as collecting preference feedback interactively can be prohibitively expensive or impractical.

Prior work on offline PbRL has introduced several algorithms with sample complexity guarantees (Zhu et al., 2023; Zhan et al., 2024a; Pace et al., 2025; Kang & Oh, 2025). However, these methods face significant practical limitations. Some are restricted to linear function approximation (Zhu et al., 2023), some require solving computationally intractable optimization problems (Zhan et al., 2024a), and others exhibit unstable performance in practice (Kang & Oh, 2025).

In particular, Zhan et al. (2024a) formulate PbRL as a distributionally robust optimization problem:

$$\hat{\pi} \in \arg\max_{\pi} \min_{r \in \mathcal{R}(\mathcal{D}),\, P \in \mathcal{P}(\mathcal{D})} V_{1,r,P}^{\pi} - V_{1,r}^{\mu},$$

where $\mathcal{R}(\mathcal{D})$ and $\mathcal{P}(\mathcal{D})$ are confidence sets for the reward and transition models. This formulation is computationally infeasible, primarily because the inner minimization over the confidence sets $\mathcal{R}(\mathcal{D})$ and $\mathcal{P}(\mathcal{D})$ requires searching over complex function classes. The overall joint optimization over policy, reward, and transition models further compounds the computational burden.

More recently, Kang & Oh (2025) proposed an actor-critic-style PbRL algorithm, APPO. By reformulating the distributionally robust optimization into a regularized optimization, their approach enables practical implementation. However, APPO achieves a suboptimal sample complexity bound of $\mathcal{O}(\varepsilon^{-4})$, which is weaker than the $\mathcal{O}(\varepsilon^{-2})$ bound of Zhan et al. (2024a). This inefficiency arises from its reliance on standard actor-critic analysis in offline RL (Xie et al., 2021; Zanette et al., 2021; Cheng et al., 2022; Nguyen-Tang & Arora, 2023), which requires bounding the cumulative conservatism bias across $T$ iterations. More importantly, APPO often exhibit from high performance variance and unstable training. Even with hyperparameter tuning, it can fail to learn effective policies (see Section 5). These issues further motivate the need for a more stable and efficient alternative for offline PbRL.

To address these challenges, we propose *Preference-based Value Optimization* (PVO), an offline PbRL algorithm that achieves both strong empirical performance and theoretical guarantees. PVO directly optimizes the value function by minimizing a novel *value alignment loss*, in conjunction with the concept of the *induced reward function*. Leveraging this formulation, we establish that PVO achieves a rate-optimal sample complexity bound of $\mathcal{O}(\varepsilon^{-2})$.

Beyond introducing PVO, we revisit APPO (Kang & Oh, 2025) and demonstrate that its variant incorporating the value alignment loss also admits a sample complexity guarantee. This finding highlights that the value alignment loss serves as a unifying principle for provably efficient PbRL, applicable to both value-based and actor-critic algorithms.

We evaluate PVO on high-dimensional continuous control benchmarks. Surprisingly, PVO consistently outperforms state-of-the-art baselines, including empirical methods that lack theoretical guarantees. It is notable that PVO exhibits robust and stable performance across various datasets, without introducing additional hyperparameters for preference learning. Furthermore, our ablation study demonstrates that replacing the standard TD loss with the value alignment loss improves the performance of RL algorithms applied to preference datasets, thereby validating the advantage of value alignment loss in PbRL. Our contributions are summarized as follows:

- **Algorithm.** We propose PVO, a simple and practical offline PbRL algorithm that achieves both strong empirical performance and theoretical guarantees. It directly optimizes the value function consistent with preference feedback by minimizing the novel value alignment loss.

- **Theoretical Guarantee.** We prove that PVO attains a rate-optimal $\mathcal{O}(\varepsilon^{-2})$ sample complexity bound (Theorem 4.1). We further show that APPO (Kang & Oh, 2025) can be modified to incorporate the value alignment loss, demonstrating that this loss is applicable to both value-based and actor–critic algorithms for provably efficient PbRL.

- **Empirical Performance.** We show that PVO outperforms state-of-the-art baselines on continuous control benchmarks. Notably, it maintains robust and stable performance across datasets where existing methods exhibit high performance variance.

- **Advantage of Value Alignment Loss.** Our ablation study reveals that replacing the standard TD loss with the value alignment loss significantly improves the performance of RL algorithms on preference datasets. This confirms that the value alignment loss provides more reliable learning signals in PbRL, where reward estimation errors are often unavoidable.

## 1.1 RELATED WORK

**Offline PbRL Theory.** In offline RL, ensuring a proper amount of conservatism in value or model estimates is essential for theoretical guarantees. The principle of conservatism still holds for offline PbRL, yet we also estimate the reward from preference feedback.

To deal with this challenge, Zhu et al. (2023) utilize pessimistic maximum likelihood estimation under linear function approximation $r(s, a) = \theta^T \phi(s, a)$. Their algorithm first constructs a confidence set for the model parameter and then performs distributionally robust policy optimization to obtain a conservative value function and corresponding policy. Zhan et al. (2024a) extended the idea to

general function classes with bounded bracketing number. They showed that bounded trajectory-level concentrability is essential for offline PbRL by establishing a lower bound. While they provide a sample complexity bound, their algorithms are computationally intractable. Recently, Kang & Oh (2025) developed a computationally efficient approach based on actor-critic style policy optimization. By framing PbRL as a two-player game between policy and reward model, they replaced the distributionally robust optimization with tractable regularized optimization. A different yet related problem setting was studied Pace et al. (2025). They developed a preference elicitation method for offline PbRL. By choosing trajectory pairs for preference queries, they eliminated the dependence on the reward concentrability coefficient.

**Empirical Studies in PbRL** Several works have explored applying deep learning techniques to preference-based reinforcement learning. A simple yet effective approach involves training a reward model on a preference dataset, then applying a standard RL algorithm using the reward signal predicted by the learned model (Christiano et al., 2017; Ibarz et al., 2018; Lee et al., 2021). The reward model is typically assumed to produce a Markovian reward as in conventional RL, although some works have considered non-Markovian rewards that depend on the entire trajectory (Kim et al., 2023; Zhang et al., 2024; Swamy et al., 2024).

A separate line of research seeks to learn value functions or policies directly from preference data, without explicitly modeling rewards. Hejna & Sadigh (2024); Hejna et al. (2024) derive a direct relationship between the preference distribution and value (or policy), and optimize the likelihood of observed preferences accordingly. An et al. (2023) use a scoring function that evaluates policy based on preference, while Kang et al. (2023) propose hindsight information matching to directly optimize the policy. Zhang et al. (2024) introduce a generative model that learns from positive/negative trajectory pairs and applies behavior cloning to the model-generated positive trajectories.

Another active area of research focuses on improving the efficiency of preference data collection. Lee et al. (2021) demonstrate that increasing trajectory diversity through unsupervised pretraining improves performance, and Liang et al. (2022) achieve a similar goal via uncertainty-based exploration. Park et al. (2022) propose data augmentation techniques tailored for PbRL, while Hejna III & Sadigh (2023) introduce a meta-learning framework for few-shot preference learning. Choi et al. (2024) extend the standard pairwise comparison setting to listwise comparisons, showing that this richer feedback provides more informative supervision.

## 2 PRELIMINARIES

**Markov Decision Processes.** We consider an episodic MDP $(\mathcal{S}, \mathcal{A}, H, P^\star, r^\star)$ with state space $\mathcal{S}$, action space $\mathcal{A}$, and horizon $H$. $P^\star = \{P_h^\star\}_{h=1}^H$ are the transition probabilities, and $r^\star = \{r_h^\star\}_{h=1}^H$ are the reward functions. For each episode, the agent starts at the initial state $s_1$[1], and then interacts with the environment for $H$ steps. At step $h \in [H]$, the agent takes action $a_h$ based on the current state $s_h$. The environment assigns reward $r_h^\star(s_h, a_h)$ and generates next state $s_{h+1}$ following the transition probability $P_h^\star(\cdot \mid s_h, a_h)$. In preference-based learning, *the agent does not observe rewards at each step, but preference feedback comparing a pair of trajectories*, as we will discuss.

The agent's strategy for taking actions is represented by policy $\pi = \{\pi_h\}_{h \in [H]}$, where $\pi_h(\cdot \mid s)$ is a probability distribution over $\mathcal{A}$. We define the state value function and the action value function of policy $\pi$ as the expected sum of rewards over an episode, following the policy $\pi$. Formally,

$$V_{h,r}^\pi(s) := \mathbb{E}_\pi \left[ \sum_{h'=h}^H r_h(s_{h'}, a_{h'}) \mid s_h = s \right], \quad Q_{h,r}^\pi := \mathbb{E}_\pi \left[ \sum_{h'=h}^H r_h(s_{h'}, a_{h'}) \mid s_h = s, a_h = a \right].$$

We write $V_{h,r^\star}^\pi$ as $V_h^\pi$ and $V_1^\pi(s_1) = V_1^\pi$ for convenience. For any policy $\pi$ and reward $r$, the Bellman equation states the relation between state and action value functions as

$$Q_{h,r}^\pi(s,a) = r_h(s,a) + P^\star V_{h+1,r}^\pi(s,a), \quad V_{h,r}^\pi(s) = Q_{h,r}^\pi(s,\pi), \quad V_{H+1}^\pi(s) = 0.$$

where we write $Pg(s,a) := \mathbb{E}_{s' \sim P(s,a)}[g(s')]$ and $f(s,\pi) := \mathbb{E}_{a \sim \pi(s)}[f(s,a)]$.

**Offline Preference-based Reinforcement Learning.** In Preference-based RL, the agent cannot observe the true reward $r^\star$ but only binary preference feedback over trajectory pairs. For a mono-

---

[1] Our analysis naturally extends with initial state distribution.

tonically increasing link function $\Phi : \mathbb{R} \mapsto [0, 1]$ with bounded $\kappa = 1/(\inf_{x \in [-R_{\max}, R_{\max}]} \Phi'(x))$, we assume the preference feedback $y^m \in \{0, 1\}$ is generated by the following model:

$$\mathbb{P}(y = 1 \mid \tau^0, \tau^1) = \mathbb{P}(\tau^1 \text{ is preferred over } \tau^0) = \Phi(r^\star(\tau^1) - r^\star(\tau^0))$$

where $r^\star(\tau) = \sum_{h=1}^H r_h^\star(s_h, a_h)$ for given trajectory $\tau = (s_1, a_1, \ldots, s_H, a_H)$. The widely used Bradely-Terry-Luce model (Bradley & Terry, 1952) is a special case of this model where $\Phi$ is set to be the sigmoid function $\sigma(x) = 1/(1 + \exp(-x))$.

We have two offline datasets: a preference dataset $\mathcal{D}_{\text{PF}} = \{(\tau^{m,0}, \tau^{m,1}, y^m)\}_{m=1}^M$ and a trajectory dataset $\mathcal{D}_{\text{TJ}} = \{(\tau^{n,0}, \tau^{n,1})\}_{n=1}^N$ where every trajectories are sampled i.i.d. by executing the reference policy $\mu$. The distinction between $\mathcal{D}_{\text{PF}}$ and $\mathcal{D}_{\text{TJ}}$ is for notational convenience. Generally, the two datasets may have common samples, e.g., we have a large $\mathcal{D}_{\text{TJ}}$ and get labels for some trajectory pairs to create $\mathcal{D}_{\text{PF}}$. Our goal is to find an $\epsilon$-optimal policy $\hat{\pi}$ with performance gap $V_{1,r^\star}^{\pi^\star}(s_1) - V_{1,r^\star}^{\hat{\pi}}(s_1) \leq \epsilon$ for an optimal policy $\pi^\star$.

**Function Approximation.** We define function classes that we use to approximate models and value functions. We have reward function class $\mathcal{R} = \mathcal{R}_1 \times \cdots \times \mathcal{R}_H \subset (\mathcal{S} \times \mathcal{A} \to [-R_{\max}, R_{\max}, ])^H$, transition function class $\mathcal{P} = \mathcal{P}_1 \times \cdots \times \mathcal{P}_H \subset (\mathcal{S} \times \mathcal{A} \to \Delta(\mathcal{S}))^H$, and function class $\mathcal{F} = \mathcal{F}_1 \times \cdots \times \mathcal{F}_H \subset (\mathcal{S} \times \mathcal{A} \to [-V_{\max}, V_{\max}, ])^H$. We denote $\pi_f$ for the greedy policy corresponding to $f$, i.e., $\pi_f(s) = \arg\max_a f(s, a)$, and define $V_f = \{V_{h,f}\}_{h \in [H]}$ as $V_f(s) = f_h(s, \pi_f(s))$ for all $s \in \mathcal{S}$. As we do not make any assumption on the structure of the function classes, the function classes can approximate complex structures such as neural networks. We define $\Pi_{\mathcal{F}}$ as the set of greedy policies corresponding to $\mathcal{F}$.

**Additional Notations.** We denote $[n] := \{1, 2, \ldots, n\}$ for $n \in \mathbb{N}$. For a given dataset $\mathcal{D}$, we use $\mathbb{E}_{x \in \mathcal{D}}[f(x)]$ to denote $\frac{1}{|\mathcal{D}|} \sum_{x \in \mathcal{D}} f(x)$. For reward function $r \in \mathcal{R}$ and trajectories $\tau^0, \tau^1$, we write $\Delta(r; \tau^0, \tau^1) = r(\tau^0) - r(\tau^1)$. For $f : \mathcal{S} \times \mathcal{A} \mapsto \mathbb{R}$, we use the notation $P_h^\pi f(s, a) := \mathbb{E}_{s' \sim P(s,a), a' \sim \pi(s')}[f(s', a')]$. The notation $f(x) \lesssim g(x)$ means that $f(x) \leq Cg(x), \forall x$ for some absolute constant $C > 0$.

## 3 ALGORITHM

In this section, we discuss how to learn a value function that is consistent with preference feedback. The key idea lies in the concept of *induced reward function* and our novel *value alignment loss*. Building on this idea, we propose Preference-based Value Optimization (PVO), a simple offline PbRL algorithm with a sample complexity guarantee. We also show that the APPO algorithm introduced by Kang & Oh (2025) can be interpreted as an actor-critic algorithm using the value alignment loss, which implies a unified framework for PbRL.

### 3.1 ALIGNING VALUE FUNCTION WITH PREFERENCE

In preference-based reinforcement learning, feedback is provided for each trajectory pair $(\tau^{m,0}, \tau^{m,1})$. To enable credit assignment, we must estimate a reward function, which can be performed by maximum likelihood estimation (MLE) over the dataset $\mathcal{D}_{\text{PF}}$. Specifically, we train a reward model $\hat{r} \in \arg\min_{r \in \mathcal{R}} \hat{L}_{\text{RW}}(r)$ where

$$\hat{L}_{\text{RW}}(r) = -\sum_{m=1}^M \log \Phi((2y^m - 1)(r(\tau^{m,1}) - r(\tau^{m,0}))) \tag{1}$$

is the negative log likelihood loss. The standard MLE concentration bound (e.g., Lemma 2 in Zhan et al. (2024a)) guarantees the following:

$$\mathbb{E}_{\tau^0, \tau^1 \sim \mu}[(\hat{r}(\tau^0) - \hat{r}(\tau^1) - r^\star(\tau^0) + r^\star(\tau^1))^2] \lesssim \frac{\kappa^2 \log(|\mathcal{R}|\delta^{-1})}{M}.$$

A key point is that this concentration bound is valid with respect to trajectory pairs $(\tau^0, \tau^1) \sim \mu$ rather than each state or transition. Consequently, the squared Bellman error used in standard RL is not compatible with PbRL. Then how can we learn value functions consistent with preference feedback? The concept of *induced reward function* plays a crucial role.

---

**Algorithm 1** PVO: Preference-based Value Optimization

---

1: **Input:** Datasets $\mathcal{D}_{\text{PF}} = \{(\tau^{m,0}, \tau^{m,1}, y^m)\}_{m=1}^M$, $\mathcal{D}_{\text{TJ}} = \{(\tau^{n,0}, \tau^{n,1})\}_{n=1}^N$
2: Estimate $\hat{r} \in \arg\min_{r \in \mathcal{R}} \hat{L}_{\text{RW}}(r)$ (1), $\hat{P}_h \in \arg\min_{P \in \mathcal{P}_h} \hat{L}_{\text{TR},h}(P)$ for all $h \in [H]$ (3)
3: Optimize $\hat{f} \in \arg\min_{f \in \mathcal{F}} \sum_{n=1}^N \left(\hat{r}_f(\tau^{n,0}) - \hat{r}_f(\tau^{n,1}) - \hat{r}(\tau^{n,0}) + \hat{r}(\tau^{n,1})\right)^2$
4: Return greedy policy $\hat{\pi} = \pi_{\hat{f}}$ such that $\pi_{\hat{f}}(s) = \arg\max_a \hat{f}(s,a)$ for all $s \in \mathcal{S}$

---

**Definition 1** (Induced Reward Function, Value Type). *For $f \in \mathcal{F}$, we define the induced reward function $r_f = \{r_{h,f}\}_{h=1}^H \in (\mathcal{S} \times \mathcal{A} \to \mathbb{R})^H$ satisfying $r_{h,f} = f_h - P_h^\star V_{h+1,f}$. Similarly, we define $\hat{r}_{h,f}$ as $\hat{r}_{h,f} = f_h - \hat{P}_h V_{h+1,f}$ where $\hat{P}$ is some transition model.*

Our *value type* induced reward is different from the *policy type* induced reward used for the analysis of actor-critic algorithms (Zanette et al., 2021; Xie et al., 2021; Nguyen-Tang & Arora, 2023; Kang & Oh, 2025) (Formal definition is presented in Definition 3). While the policy type definition is rooted in the Bellman equation $Q_h^\pi = r_h^\star + P_h^\star V_{h+1}^\pi$ with respect to some policy $\pi$, our value type definition is inspired by the *Bellman optimality equation* $Q_h^{\pi^*}(s,a) = r_h^\star(s,a) + \mathbb{E}_{s' \sim P_h^\star}[\max_{a'} Q_{h+1}^{\pi^*}(s',a')]$. The value type induced reward enables us to directly optimize the value function without actor-critic iterations as in Kang & Oh (2025), leading to more stable and efficient learning, as we will see in Section 5.

Equipped with the definition of induced reward function, we introduce the *value alignment loss*, a simple loss function for consistent value learning:

$$\hat{L}_{\text{VA}}(r_f, \hat{r}) = \sum_{n=1}^N \left(r_f(\tau^{n,0}) - r_f(\tau^{n,1}) - \hat{r}(\tau^{n,0}) + \hat{r}(\tau^{n,1})\right)^2. \tag{2}$$

At first glance, $\hat{L}_{\text{VA}}$ can be viewed as the squared trajectory-level error between the induced reward function $r_f$ and the reward model $\hat{r}$. While this view is valid, a more structured interpretation is obtained by Definition 1:

$$\hat{L}_{\text{VA}}(r_f, \hat{r}) = \sum_{n=1}^N \left(\sum_{h=1}^H (f_h - \hat{r}_h - P_h^\star V_{h+1,f})(s_h^{n,0}, a_h^{n,0}) - \sum_{h=1}^H (f_h - \hat{r}_h - P_h^\star V_{h+1,f})(s_h^{n,1}, a_h^{n,1})\right)^2.$$

This expression reveals that $\hat{L}_{\text{VA}}$ represents the difference in the cumulative Bellman errors of $f$ between a pair of trajectories. Minimizing $\hat{L}_{\text{VA}}$ encourages $f$ to be Bellman-consistent with respect to $\hat{r}$, thus aligning it with the preference data. We therefore refer to $\hat{L}_{\text{VA}}$ as value alignment loss.

### 3.2 PREFERENCE-BASED VALUE OPTIMIZATION

We present PVO, a direct application of the value alignment loss with Definition 1. The pseudo-code is presented in Algorithm 1.

**Model Learning.** Our algorithm consists of two phases: model learning (Line 2) and value optimization (Line 3). In the model learning phase, we train reward and transition models via maximum likelihood estimation. Formally, we compute $\hat{r} = \arg\min_{r \in \mathcal{R}} \hat{L}_{\text{RW}}(r)$ (1) and $\hat{P}_h \in \arg\min_{P \in \mathcal{P}_h} \hat{L}_{\text{TR},h}(P)$ where

$$\hat{L}_{\text{TR},h}(P) = -\sum_{n=1}^N \sum_{j \in \{0,1\}} \log P(s_{h+1}^{n,j} \mid s_h^{n,j}, a_h^{n,j}). \tag{3}$$

**Value Optimization.** In the value optimization phase, we minimize the value alignment loss:

$$\hat{f} \in \arg\min_{f \in \mathcal{F}} \underbrace{\sum_{n=1}^N \left(\hat{r}_f(\tau^{n,0}) - \hat{r}_f(\tau^{n,1}) - \hat{r}(\tau^{n,0}) + \hat{r}(\tau^{n,1})\right)^2}_{\text{value alignment loss } \hat{L}_{\text{VA}}(\hat{r}_f, \hat{r})}. \tag{4}$$

As we discussed, this facilitates the consistency of the value function $\hat{f}$ with respect to the reward estimation, $\hat{r}$. Since (4) is unconstrained, we can easily implement it using neural networks and off-the-shelf gradient-based optimizers. We present a practical deep RL implementation in Section 5.

## 3.3 REVISITING APPO WITH VALUE ALIGNMENT LOSS

The APPO algorithm (Kang & Oh, 2025) alternates between policy updates of the form $\pi_h^{t+1}(a \mid s) \propto \pi_h^t(a \mid s) \exp(\eta f_h^t(s, a))$ and value function optimization:

$$f^t \in \arg\min_{f \in \mathcal{F}} \left( \lambda \sum_{n=1}^{N} \sum_{h=1}^{H} \left[ f_h(s_h^{n,0}, \pi_h^t(s_h^{n,0})) - f_h(s_h^{n,0}, a_h^{n,0}) \right] + \hat{\mathcal{E}}(f) \right)$$

where $\hat{\mathcal{E}}(f) = \sum_{n=1}^{N} \left| \hat{r}_f^{\pi^t}(\tau^{n,0}) - \hat{r}_f^{\pi^t}(\tau^{n,1}) - \hat{r}(\tau^{n,0}) + \hat{r}(\tau^{n,1}) \right|$ is the $\ell_1$ loss between the policy type induced reward $\hat{r}_{h,f}^{\pi^t}$ and the reward model $\hat{r}$. This can be viewed as an $\ell_1$ variant of the value alignment loss $\hat{L}_{\text{VA}}$, leading to a natural question:

**Question**: *Can we still guarantee a sample complexity bound if $\hat{\mathcal{E}}(f)$ is replaced by $\hat{L}_{VA}$ in APPO?*

We answer this affirmatively: if APPO is modified to use value alignment loss as in Algorithm 2, it enjoys a sample complexity guarantee (Theorem B.1). The detailed discussion and analysis is presented in Appendix B. This suggests that the combination of the value alignment loss and the induced reward function provides a unified framework applicable to both value optimization and actor-critic methods.

## 3.4 PRACTICAL IMPLEMENTATION

PVO can be practically implemented with neural networks empowered by off-the-shelf deep learning methods. We adapt PVO to the standard discounted MDP setting for deep PbRL (Christiano et al., 2017), where we have preference feedback for trajectory segment pairs of length $L$.

**Reward Learning.** Since the value optimization objective 4 uses a reward model $\hat{r}$, we train a reward model based on the preference dataset $\mathcal{D}_{\text{PF}}$. We train $\hat{r}$ by maximizing log likelihood $\hat{L}_{\text{RW}}(\mathcal{D}_{\text{PF}})$, yet it is possible to employ other advanced techniques for preference learning (Park et al., 2022; Shin et al., 2023; Hwang et al., 2024; Choi et al., 2024). We note that reward model training requires minimal computational cost: In our experiments, training a reward model with 1000 preference samples takes less than a minute, while value and policy learning take more than 2 hours for all algorithms.

**Value Optimization.** To implement the value optimization (Line 3 in Algorithm 1) with deep neural networks, we parameterize Q and V functions separately. The V function is trained via expectile regression (Kostrikov et al., 2022):

$$L_V(\mathcal{D}_{\text{TJ}}) = \mathbb{E}_{(s,a) \in \mathcal{D}_{\text{TJ}}} \left[ L_2^\tau(Q(s,a) - V(s)) \right] \tag{5}$$

where $L_2^\tau(u) = |\tau - \mathbb{1}\{u < 0\}| u^2$. The optimization objective for the Q function is

$$L_Q(\mathcal{D}_{\text{TJ}}) = \mathbb{E}_{(\tau^0, \tau^1) \in \mathcal{D}_{\text{TJ}}} \left[ \left( r_{Q,V}(\tau^0) - r_{Q,V}(\tau^1) - \hat{r}(\tau^0) + \hat{r}(\tau^1) \right)^2 \right]. \tag{6}$$

where $r_{Q,V}(\tau) = \sum_{l=1}^{L} (Q(s_l, a_l) - \gamma V(s_{h+l}))$ and $\pi(s)$ is an action sampled from $\pi(\cdot \mid s)$. We use $V(s_{l+1})$ instead of $\hat{P}V(s_l, a_l)$, eliminating the need for training a transition model. This approximation leads to good empirical performance, as shown in the experimental results.

Finally, the policy is extracted via advantage weighted regression (Peng et al., 2019):

$$L_\pi(\mathcal{D}_{\text{TJ}}) = \mathbb{E}_{(s,a) \in \mathcal{D}_{\text{TJ}}} \left[ \exp(\beta(Q(s,a) - V(s))) \log \pi(a \mid s) \right]. \tag{7}$$

## 4 THEORETICAL ANALYSIS

This section presents the sample complexity analysis of our proposed algorithms, PVO. Our analysis is based on some standard assumptions in the PbRL literature. First, we assume that the function classes are realizable (Chen et al., 2023; Zhan et al., 2024a; Pace et al., 2025; Kang & Oh, 2025).

**Assumption 1** (Reward Function Class). *The reward function class is realizable, i.e., $r^\star \in \mathcal{R}$. For every $r \in \mathcal{R}$ and trajectory $\tau$, it holds that $|r(\tau)| \leq R_{max}$.*

**Assumption 2** (Transition Function Class). *The transition function class is realizable, i.e., $P^\star \in \mathcal{P}$.*

**Assumption 3** (Value Function Class). *For any policy $\pi$, we have $Q^\pi \in \mathcal{F}$. In addition, $|f_h(s,a)| \leq V_{max}$ for all $f \in \mathcal{F}$, $h \in [H]$, and $(s,a) \in \mathcal{S} \times \mathcal{A}$.*

We define a PbRL version of uniform concentrability coefficient Munos & Szepesvári (2008); Chen & Jiang (2019). Note that is defined with respect to trajectory density instead of state-action density.

**Definition 2** (PbRL Uniform Concentrability).

$$\mathcal{C}_\mu(\mathcal{F}) = \sup_{\pi \in \Pi_\mathcal{F}} \sup_{f \in \mathcal{F}} \frac{|\mathbb{E}_{\tau^0 \sim \pi, \tau^1 \sim \mu}[r_f(\tau^0) - r_f(\tau^1) - r^\star(\tau^0) + r^\star(\tau^1)]|}{\sqrt{\mathbb{E}_{\tau^0, \tau^1 \sim \mu}[(r_f(\tau^0) - r_f(\tau^1) - r^\star(\tau^0) + r^\star(\tau^1))^2]}}$$

Now we present the sample complexity bounds. The proofs are presented in Section C. Note that our analysis naturally extends to infinite function classes using the standard covering number argument.

**Theorem 4.1** (Sample complexity of PVO). *Suppose Assumptions 1, 2, and 3 hold. With probability at least $1 - \delta$, Algorithm 1 achieves an $\varepsilon$-optimal policy with*

$$M = \mathcal{O}\Big(\frac{\mathcal{C}_\mu(\mathcal{F})^2 \kappa^2 \log(|\mathcal{R}|\delta^{-1}))}{\varepsilon^2}\Big), N = \mathcal{O}\left(\frac{\mathcal{C}_\mu(\mathcal{F})^2 V_{max}^2 H^2 (\log(|\mathcal{P}|H\delta^{-1}) + \log(|\mathcal{F}|\delta^{-1})))}{\varepsilon^2}\right).$$

Compared to the sample complexity bounds of FREEHAND-transition (Zhan et al., 2024a), the bound in Theorem 4.1 is looser since the uniform concentrability $\mathcal{C}_\mu$ is a stronger condition than the single-policy concentrability (Zhan et al., 2024a; Kang & Oh, 2025). However, the sample complexity bound of PVO still achieves rate-optimal $\mathcal{O}\left(\varepsilon^{-2}\right)$ for both $M$ and $N$. Moreover, the sharp bounds of FREEHAND-transition comes at the cost: FREEHAND-transition is computational intractable due to its reliance on a distributionally robust optimization oracle. Compared with APPO (Kang & Oh, 2025), PVO's bound on $N$ is sharper than $\mathcal{O}\left(\varepsilon^{-4}\right)$ bound of APPO, but APPO relies on the weaker single-policy concentrability assumption. We note that APPO also has empirical limitations such as unstable performance and additional hyperparameters as explained in Section 5. Therefore, the sample complexity bound of PVO reveals a trade-off: despite having a weaker bound, PVO offers significant advantages in terms of practical implementation and empirical performance.

## 5 EXPERIMENTS

In this section, we evaluate PVO in continuous control benchmarks with elaborate ablation study.

### 5.1 EXPERIMENTAL SETUP

We evaluate PVO on the Meta-World (Yu et al., 2020) and DMControl (Tassa et al., 2018) datasets from Choi et al. (2024). They are widely used continuous control benchmarks with high-dimensional state spaces. We mainly use Meta-World datasets for evaluation, and the results with the DMControl dataset is presented in Appendix F. We follow the experimental setup of Choi et al. (2024) and Kang & Oh (2025). The preference dataset consists of pairs of randomly sampled trajectory segments of length 25, and the preference labels are generated based on the ground truth return of segments. We measure algorithm performance using the success rate for Meta-World tasks and the episodic return for DMControl tasks.

We consider the following baselines: (1) IQL (Kostrikov et al., 2022) with a learned reward model is a simple yet strong baseline that has been widely adopted in previous studies (Kim et al., 2023; An et al., 2023; Hejna & Sadigh, 2024; Hejna et al., 2024; Choi et al., 2024); (2) APPO (Kang & Oh, 2025) is a provably efficient algorithm based on adversarial training; (3) Preference Transformer (PT) (Kim et al., 2023) utilizes the Transformer (Vaswani, 2017) architecture for sequential reward modeling; (4) DPPO (An et al., 2023) directly optimizes policy with preference score metric; (5) IPL (Hejna & Sadigh, 2024) optimizes policy by maximizing the likelihood of observed preference data. Further details on the setup are presented in Appendix G.

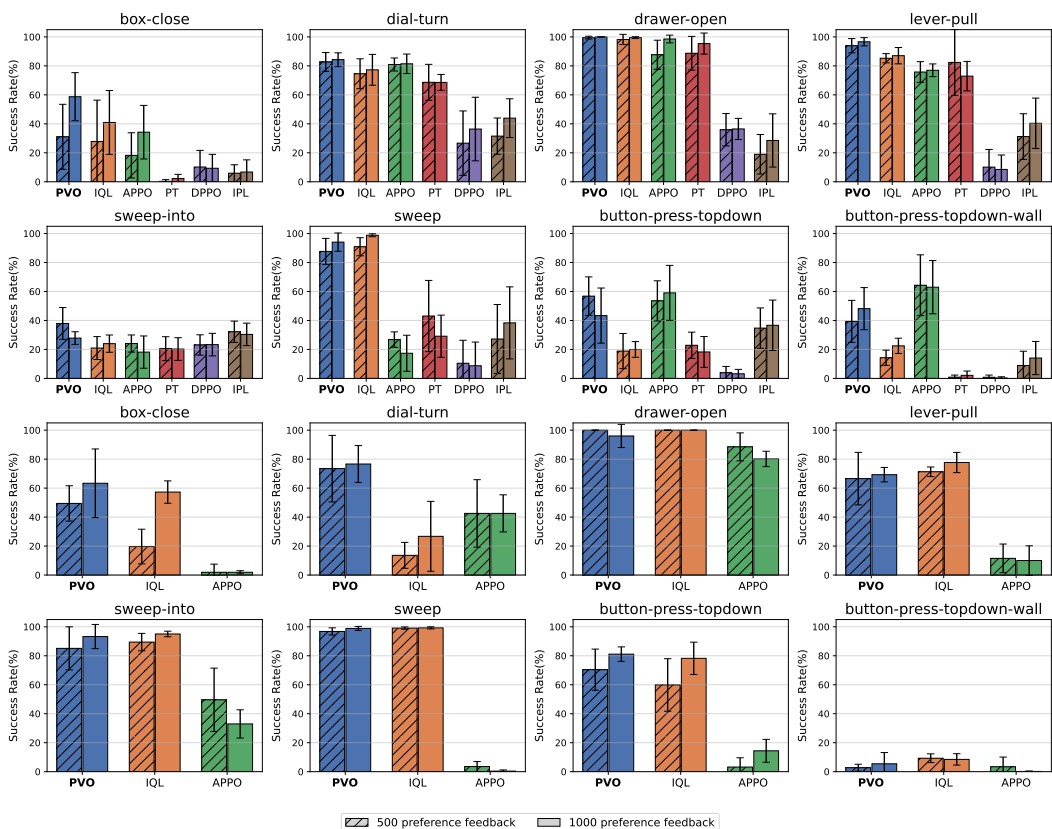

Figure 1: (Top two rows) Performance on Meta-World `medium-replay` datasets and (Bottom two rows) `medium-expert` datasets, measured by success rate. For `medium-replay` datasets, we include the results of PT, DPPO, and IPL from Choi et al. (2024). For the `medium-expert` datasets, we evaluate the top three algorithms from the `medium-replay` datasets. Each plot shows the mean and standard deviation over five random seeds.

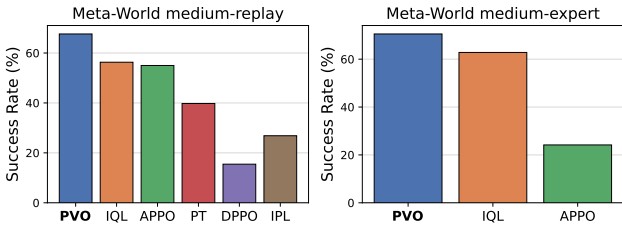

Figure 2: Average performance on Meta-World `medium-replay` and `medium-expert` datasets.

## 5.2 EVALUATION ON PREFERENCE DATASETS

Figure 1 presents the performance of algorithms on Meta-World datasets, and Figure 2 summarizes the overall results. PVO consistently outperforms baseline methods across diverse environments. In particular, while baselines often exhibit high variance across datasets, PVO maintains robust performance. For example, IQL performs comparably to PVO on the `medium-replay` sweep dataset, but fails to learn on the `medium-replay` button-press-topdown dataset. This instability arises from the nature of preference feedback, where agents rely on potentially misspecified reward estimation. The stable performance of PVO indicates greater robustness to such reward model errors, which is a significant advantage in the PbRL setting.

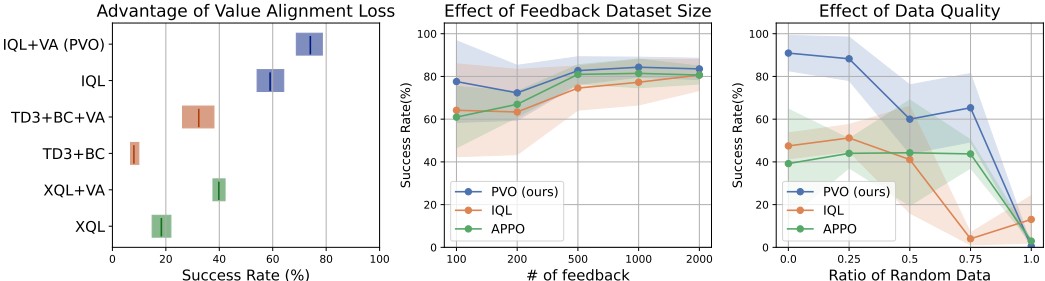

Figure 3: (Left) Overall performance of IQL, XQL, and TD3+BC with and without value alignment loss (+VA indicates the modified version using the value alignment loss), aggregated by interquartile mean (Agarwal et al., 2021). (Middle) Performance of algorithms on Meta-World `medium-replay` dial-turn dataset, with varying number of preference feedback. (Right) Performance of algorithms on mixture datasets of random and expert trajectories, with varying proportions of random trajectories.

Moreover, PVO introduces no additional hyperparameters for preference learning, requiring exactly the same set of hyperparameters as IQL. This stands in contrast to existing PbRL methods that depend on extra hyperparameters, such as the conservatism parameter in APPO, the smoothness and conservatism regularizers in DPPO, and the regularization parameter in IPL.

## 5.3 ABLATION STUDY

**Advantage of Value Alignment Loss.**

The improvement of PVO over baselines stems from the value alignment loss (6). To isolate its effect, we implemented additional baselines: TD3+BC (Fujimoto & Gu, 2021) (actor–critic) and XQL (Garg et al., 2023) (value-based). We compared their original versions, which use the standard TD loss, with variants that employ the value alignment loss.

The left plot of Figure 2 reports results on the Meta-World `medium-replay` datasets (8 tasks) with 1000 preference feedback. For both TD3+BC and XQL, replacing the TD loss with the value alignment loss yields substantial improvements. A similar pattern appears in the gap between PVO and IQL, which share identical network architectures and expectile regression but differ in the use of the value alignment loss. These results confirm that the value alignment loss provides a more informative learning signal than the standard TD loss.

From a theoretical perspective, we hypothesize that its advantage lies in mitigating the propagation of reward model errors. Unlike the TD loss, which can amplify such errors through Bellman backups, the value alignment loss distributes errors at the trajectory level, smoothing their impact on value estimation. This property may explain its empirical effectiveness in PbRL, where reward model misspecification is often inevitable.

**Effect of Preference Dataset Size and Data Quality.**

We next study sensitivity to the amount of preference feedback. The middle plot of Figure 3 shows that PVO achieves effective learning with as few as about 100 preference samples, exhibiting minimal performance degradation.

We also examine the effect of data quality. On the Meta-World dial-turn task, we constructed mixture datasets by combining expert and random trajectories with varying proportions $r \in \{0, 0.25, 0.5, 0.75, 1\}$. Here, $r = 0$ corresponds to an expert dataset, while $r = 1$ denotes a fully random dataset. The result is presented in the right plot of Figure 3. As expected, performance declines for all algorithms as $r$ increases, since a larger proportion of random trajectories reduces the amount of high-quality data available for learning. Importantly, PVO consistently maintains superiority across all mixture settings, demonstrating robustness even when the dataset substantially diverges from the optimal policy distribution.

## REPRODUCIBILITY

The experimental details are described in Section 5 and Section G, including hyperparameters, neural network architecture, and dataset information. The code used to run experiments can be found in the supplementary material where the README file explains how to configure training environment and execute scripts. The Meta-World `medium-replay` datasets and DMControl datasets are available in the official repository of Choi et al. (2024). The Meta-World `medium-expert` datasets are generated using the script provided in the official repository of Hejna & Sadigh (2024).

## ACKNOWLEDGEMENTS

This work was supported by the National Research Foundation of Korea (NRF) grant and the Institute of Information & communications Technology Planning & Evaluation (IITP) grant both funded by the Korea government (MSIT) (No. RS-2022-NR071853, RS-2023-00222663, RS-2025-25463302) and by AI-Bio Research Grant through Seoul National University. Hyungkyu Kang was supported by an internship at Upstage.

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

## A ADDITIONAL RELATED WORK

**Problem Settings in Preference-based RL.** Preference-based RL, sometimes called reinforcement learning from human feedback (RLHF), involves learning from preference feedback rather than explicit reward signals. Broadly, existing work in PbRL can be categorized into two lines: The first—which we focus on in this paper—studies learning from preference feedback in general stochastic MDPs (e.g. (Novoseller et al., 2020; Christiano et al., 2017)); The second line concentrates on deterministic MDPs or bandit problems, typically in the context of large language models (e.g. (Rafailov et al., 2024; Xiong et al., 2024; Rosset et al., 2024; Xie et al., 2025)). The latter setting often considers fine-tuning a pretrained policy using preference feedback, and incorporates regularization toward a pre-trained policy. In this work, we focus on the former setting and review related work accordingly.

**Online PbRL Theory.** The theoretical analysis on online PbRL has emerged from the dueling bandit problem (Yue et al., 2012), where the agent makes sequential decisions based on preferences between selected actions. One of the earliest approaches was made by Novoseller et al. (2020), who establish an asymptotic Bayesian regret bound for a posterior sampling algorithm in tabular MDP. Xu et al. (2020) combine a reward-free exploration strategy and dueling bandit subroutines, offering a finite-time sample complexity bound. Several works have studied PbRL with linear models. Saha et al. (2023) propose a bandit-like algorithm that treats policy as action, and Zhan et al. (2024b) develop a reward-agnostic experimental design algorithm. Wu & Sun (2024) utilize posterior sampling techniques to prove a worst-case regret bound for the linear setting, and a Bayesian regret for general function classes. Beyond linear settings, preference learning with general function approximation has gained attention. For instance, Chen et al. (2022) design an algorithm using exploration bonus that achieves a regret bound dependent on Eluder dimension (Russo & Van Roy, 2013). Wang et al. (2023) propose a reward learning framework that solves PbRL when augmented with standard RL algorithms. Du et al. (2024) analyze policy optimization algorithms for PbRL, under linear and neural function approximation. Chen et al. (2023) and Zhao et al. (2024) study risk-aware PbRL using the conditional value-at-risk (CVaR) objective (Artzner, 1997). Another angle was explored by Swamy et al. (2024), who formulated the PbRL problem as a two-player zero-sum game over policies, thereby generalizing to arbitrary reward representations.

## B APPLYING VALUE ALIGNMENT LOSS TO APPO

---

**Algorithm 2** PAC: Preference-based Actor-Critic (A variant of APPO (Kang & Oh, 2025))

---

1: **Input:** Datasets $\mathcal{D}_{\text{PF}}$, $\mathcal{D}_{\text{TJ}}$, constants $\eta, \lambda$, Initial policy $\pi_h^1 = \text{Unif}(\mathcal{A})$ for all $h \in [H]$
2: Estimate $\hat{r} \in \arg\min_{r \in \mathcal{R}} \hat{L}_{\text{RW}}(r)$ (1), $\hat{P}_h \in \arg\min_{P \in \mathcal{P}_h} \hat{L}_{\text{TR},h}(P)$ for all $h \in [H]$ (3)
3: **for** $t = 1, \cdots, T$ **do**
4:      $f^t \in \underset{f \in \mathcal{F}}{\arg\min} \left( \lambda \Big( f_1(s_1, \pi^t) - \frac{1}{N} \sum_{n=1}^N \hat{r}_f^{\pi^t}(\tau^{n,0}) \Big) + \hat{L}_{\text{VA}}(\hat{r}_f^{\pi^t}, \hat{r}) \right)$
5:      Update policy $\pi_h^{t+1}(a \mid s) \propto \pi_h^t(a \mid s) \exp(\eta f_h^t(s,a))$ for $h \in [H]$
6: **end for**
7: Return $\hat{\pi} = \frac{1}{T} \sum_{t=1}^T \pi^t$

---

In this section, we discuss how value alignment loss can be applied to APPO. We begin by formally defining the policy type induced reward function.

**Definition 3** (Induced Reward Function, Policy Type (Zanette et al., 2021)). *For $f \in \mathcal{F}$ and $\pi \in \Pi$, the induced reward function $r_f^\pi = \{r_{h,f}^\pi\}_{h=1}^H \in (\mathcal{S} \times \mathcal{A} \to \mathbb{R})^H$ is defined as $r_{h,f}^\pi = f_h - P_h^{\star,\pi} f_{h+1}$. Similarly, we define $\hat{r}_{h,f}^\pi$ as $\hat{r}_{h,f}^\pi = f_h - \hat{P}_h^\pi f_{h+1}$ where $\hat{P}$ is some transition model.*

We can naturally modify APPO to utilize value alignment loss with policy type induced reward. The pseudo-code is presented in Algorithm 2. In the modified algorithm PAC, the $\ell_1$ loss between the policy type induced reward $\hat{r}_{h,f}^{\pi^t}$ and the reward model $\hat{r}$ is replaced with value alignment loss $\hat{L}_{\text{VA}}(\hat{r}_{h,f}^{\pi^t}, \hat{r})$.

Now we theoretically analyze `PAC`. Following Zhan et al. (2024a); Kang & Oh (2025), we define the PbRL version of single-policy concentrability (Xie et al., 2021; Uehara & Sun, 2021).

**Definition 4** (PbRL Single Policy Concentrability (Zhan et al., 2024a; Kang & Oh, 2025)[2])**.**

$$\mathcal{C}^{\star}_{\mu,T}(\mathcal{F}) = \sup_{f \in \mathcal{F}, \pi \in \Pi^{soft}_{\mathcal{F},T}} \frac{|\mathbb{E}_{\tau^0 \sim \pi^{\star}, \tau^1 \sim \mu}[r_f^{\pi}(\tau^0) - r_f^{\pi}(\tau^1) - r^{\star}(\tau^0) + r^{\star}(\tau^1)]|}{\sqrt{\mathbb{E}_{\tau^0, \tau^1 \sim \mu}[(r_f(\tau^0) - r_f(\tau^1) - r^{\star}(\tau^0) + r^{\star}(\tau^1))^2]}}$$

*where* $\Pi^{soft}_{\mathcal{F},T} = \{\pi = \{\pi_h\}_{h=1}^H \mid \pi_h \propto \exp(\eta \sum_{i=1}^t f_h^i) \ \forall h \in [H], f^1, \ldots, f^t \in \mathcal{F}, t \in [T], \eta > 0\}$ *is the set of softmax policies.*

The single policy concentrability is bounded by the trajectory density ratio $C_{TR} = \sup_{\tau} \frac{d^{\pi^{\star}}(\tau)}{d^{\mu}(\tau)}$, and it is known that the sample complexity of offline PbRL is lower bounded by $C_{TR}$ (Zhan et al., 2024a). We have the following sample complexity bound for `PAC`:

**Theorem B.1** (Sample complexity of `PAC`)**.** *Suppose Assumptions 1, 2, and 3 hold. For properly set $\eta$ and $\lambda$, with probability at least $1 - \delta$, Algorithm 2 achieves an $\varepsilon$-optimal policy with $T = \frac{V_{max}^2 H^2 \log |\mathcal{A}|}{\varepsilon^2}$,*

$$M = \mathcal{O}\left(\frac{\mathcal{C}^{\star}_{\mu,T}(\mathcal{F})^2 \kappa^2 \log(|\mathcal{R}|\delta^{-1}))}{\varepsilon^2}\right), \mathcal{O}\left(\frac{\mathcal{C}^{\star}_{\mu,T}(\mathcal{F})^2 V_{max}^2 H^2 (\log(|\mathcal{P}|H\delta^{-1}) + T \log(|\mathcal{F}|\delta^{-1})))}{\varepsilon^2}\right).$$

The sample complexity bound for $M$ matches that of APPO, while the bound for $N$ incurs an additional dependence on $\mathcal{C}^{\star}_{\mu,T}(\mathcal{F})$. This dependency arises from the use of the quadratic value alignment loss, which requires a variant of the decoupling argument (e.g. (Foster et al., 2021; Dann et al., 2021; Jin et al., 2021)) rather than the direct suboptimality decomposition in Kang & Oh (2025). Theorem B.1 is significant in that it demonstrates the value alignment loss can be used in actor-critic algorithms. Together with the analysis of `PVO`, this suggests that the *value alignment loss provides a unifying framework for provably efficient PbRL*, applicable to both value-based and actor-critic methods.

## C    DETAILED PROOFS

We present the proofs omitted in Section 4.

### C.1    PROOF OF THEOREM 4.1

First, we prove Theorem 4.1. As we discussed, our `PVO` is a new type of algorithm that directly optimizes the value function without actor-critic iteration or policy optimization oracle. Therefore, our proof relies on a novel suboptimality decomposition utilizing the greedy property of $\hat{\pi} = \pi_{\hat{f}}$.

*Proof of Theorem 4.1.* By Lemma C.1, we have the following regret decomposition:

$$V_1^{\pi^{\star}}(s_1) - V_1^{\hat{\pi}}(s_1)$$
$$\leq V_1^{\pi^{\star}}(s_1) - V_{1,r_{\hat{f}}}^{\pi^{\star}}(s_1) + V_{1,r_{\hat{f}}}^{\hat{\pi}}(s_1) - V_1^{\hat{\pi}}(s_1)$$
$$= \left(V_1^{\pi^{\star}}(s_1) - V_{1,r_{\hat{f}}}^{\pi^{\star}}(s_1) + V_{1,r_{\hat{f}}}^{\mu}(s_1) - V_1^{\mu}(s_1)\right) + \left(V_{1,r_{\hat{f}}}^{\hat{\pi}}(s_1) - V_1^{\hat{\pi}}(s_1) - V_{1,r_{\hat{f}}}^{\mu}(s_1) + V_1^{\mu}(s_1)\right)$$
$$= \underbrace{\mathbb{E}_{\tau^0 \sim \pi^{\star}, \tau^1 \sim \mu}[\Delta(r^{\star}; \tau^0, \tau^1) - \Delta(r_{\hat{f}}; \tau^0, \tau^1)]}_{(I)} + \underbrace{\mathbb{E}_{\tau^0 \sim \hat{\pi}, \tau^1 \sim \mu}[\Delta(r_{\hat{f}}; \tau^0, \tau^1) - \Delta(r^{\star}; \tau^0, \tau^1)]}_{(II)}$$

The terms (I) and (II) represent the error of the induced reward function $r_f$, under joint distributions $(\pi^{\star}, \mu)$ and $(\hat{\pi}, \mu)$, respectively. By Definition 2, each term is bounded by

$$(I), (II) \leq \sqrt{\mathcal{C}_{\mu}(\mathcal{F})^2 \mathbb{E}_{\tau^0, \tau^1 \sim \mu}[(\Delta(r^{\star}; \tau^0, \tau^1) - \Delta(r_{\hat{f}}; \tau^0, \tau^1))^2]}.$$

---

[2]We present the definition that APPO (Kang & Oh, 2025) implicitly relies on—although it is not explicitly stated in the paper—which is slightly stronger than the one used in Zhan et al. (2024a).

Therefore, it is left to bound the term $\mathbb{E}_{\tau^0,\tau^1\sim\mu}[(\Delta(r^\star;\tau^0,\tau^1) - \Delta(r_{\hat{f}};\tau^0,\tau^1))^2]$, which can be interpreted as the population version of our value alignment loss $\hat{L}_{\mathrm{VA}}$. Lemma C.2 provides the bound, thus we finally have

$$V_1^{\pi^\star}(s_1) - V_1^{\hat{\pi}}(s_1) \leq \mathcal{O}\left(\sqrt{\mathcal{C}_\mu(\mathcal{F})^2\left(\frac{\kappa^2\log(|\mathcal{R}|\delta^{-1})}{M} + \frac{H^2V_{\max}^2[\log(|\mathcal{P}|H\delta^{-1}) + \log(|\mathcal{F}|\delta^{-1})]}{N}\right)}\right).$$

This implies the sample complexity presented in Theorem 4.1. $\qquad\square$

**Lemma C.1.** *For any $f \in \mathcal{F}$ and any policy $\pi$, it holds that $V_{1,r_f}^\pi(s_1) \leq V_{1,f}(s_1) = V_{1,r_f}^{\pi_f}(s_1)$.*

*Proof.* For $h \in [H+1]$, we have $f_h(s,\pi) \leq f_h(s,\pi_f) = V_{h,f}(s)$ for all $s \in \mathcal{S}$ by definition (we set $f_{H+1} = 0$). Therefore, we have

$$\begin{aligned}
V_{1,r_f}^\pi &= \mathbb{E}_\pi\left[\sum_{h=1}^H (f_h - P_h^\star V_{h+1,f})(s_h,a_h)\right] \\
&= \mathbb{E}_\pi\left[\sum_{h=1}^H f_h(s_h,\pi) - \mathbb{E}_{s_{h+1}\sim P_h^\star(s_h,a_h)}[V_{h+1,f}(s_{h+1})]\right] \\
&= \mathbb{E}_\pi\left[\sum_{h=1}^H f_h(s_h,\pi) - \mathbb{E}_{s_{h+1}\sim P_h^\star(s_h,a_h)}[f_{h+1}(s_{h+1},\pi_f)]\right] \\
&\leq \mathbb{E}_\pi\left[\sum_{h=1}^H f_h(s_h,\pi) - \mathbb{E}_{s_{h+1}\sim P_h^\star(s_h,a_h)}[f_{h+1}(s_{h+1},\pi)]\right] \\
&= f_1(s_1,\pi) \leq V_{1,f}(s_1)
\end{aligned}$$

where we used telescoping sum in the second last step. Now applying a similar argument in reverse order, we further have

$$\begin{aligned}
V_{1,f}(s_1) &= f_1(s_1,\pi_f) \\
&= \sum_{h=1}^H \mathbb{E}_{\pi_f}[f_h(s_h,a_h) - \mathbb{E}_{s_{h+1}\sim P_h^\star(s_h,a_h)}[f_{h+1}(s_{h+1},\pi_f)]] \\
&= \mathbb{E}_{\pi_f}[\sum_{h=1}^H (f_h - P_h^\star V_{h+1,f})(s_h,a_h)] = V_{1,r_f}^{\pi_f}
\end{aligned}$$

$\qquad\square$

**Lemma C.2.** *With probability at least $1 - \delta$, we have*

$$\mathbb{E}_{\tau^0,\tau^1\sim\mu}[(\Delta(r^\star;\tau^0,\tau^1) - \Delta(r_{\hat{f}};\tau^0,\tau^1))^2]$$
$$\lesssim \frac{\kappa^2\log(|\mathcal{R}|\delta^{-1})}{M} + \frac{H^2V_{max}^2[\log(|\mathcal{P}|H\delta^{-1}) + \log(|\mathcal{F}|\delta^{-1})]}{N}$$

*Proof.* Lemma C.3 and Lemma D.5 implies that

$$\begin{aligned}
&N\mathbb{E}_{\tau^0,\tau^1\sim\mu}[(\Delta(r_{\hat{f}};\tau^0,\tau^1) - \Delta(\hat{r};\tau^0,\tau^1))^2] \\
&\leq 2\hat{L}_{\mathrm{VA}}(r_{\hat{f}},\hat{r}) + 16H^2V_{\max}^2\log(|\mathcal{F}|\delta^{-1}) \\
&\leq 4\hat{L}_{\mathrm{VA}}(\hat{r}_{\hat{f}},\hat{r}) + 8HV_{\max}^2\sum_{n=1}^N\sum_{h=1}^H\sum_{j\in\{0,1\}}\left\|P_h^\star - \hat{P}_h\right\|_1^2(s_h^{n,j},a_h^{n,j}) + 16H^2V_{\max}^2\log(|\mathcal{F}|\delta^{-1}) \\
&\leq 4\hat{L}_{\mathrm{VA}}(\hat{r}_{\hat{f}},\hat{r}) + 8c_2H^2V_{\max}^2\log(|\mathcal{P}|H\delta^{-1}) + 16H^2V_{\max}^2\log(|\mathcal{F}|\delta^{-1})
\end{aligned}$$

and similarly,

$$N\mathbb{E}_{\tau^0,\tau^1\sim\mu}[(\Delta(r_{Q^{\pi^\star}};\tau^0,\tau^1)-\Delta(\hat{r};\tau^0,\tau^1))^2]$$

$$\geq \frac{2}{3}\hat{L}_{\mathsf{VA}}(r_{Q^{\pi^\star}},\hat{r})-\frac{16}{3}H^2V_{\max}^2\log(|\mathcal{F}|\delta^{-1})$$

$$\geq \frac{1}{3}\hat{L}_{\mathsf{VA}}(\hat{r}_{Q^{\pi^\star}},\hat{r})-\frac{4}{3}HV_{\max}^2\sum_{n=1}^{N}\sum_{h=1}^{H}\sum_{j\in\{0,1\}}\left\|P_h^\star-\hat{P}_h\right\|_1^2(s_h^{n,j},a_h^{n,j})-\frac{16}{3}H^2V_{\max}^2\log(|\mathcal{F}|\delta^{-1})$$

$$\geq \frac{1}{3}\hat{L}_{\mathsf{VA}}(\hat{r}_{Q^{\pi^\star}},\hat{r})-\frac{4}{3}c_2H^2V_{\max}^2\log(|\mathcal{P}|H\delta^{-1})-\frac{16}{3}H^2V_{\max}^2\log(|\mathcal{F}|\delta^{-1})$$

On the other hand, the optimality of $\hat{f}$ (Line 3 in Algorithm 1) implies that

$$\hat{L}_{\mathsf{VA}}(\hat{r}_{\hat{f}},\hat{r})\leq \hat{L}_{\mathsf{VA}}(\hat{r}_{Q^{\pi^\star}},\hat{r}).$$

Combining the results, we have

$$N\mathbb{E}_{\tau^0,\tau^1\sim\mu}[(\Delta(r_{\hat{f}};\tau^0,\tau^1)-\Delta(\hat{r};\tau^0,\tau^1))^2]$$

$$\leq 4\hat{L}_{\mathsf{VA}}(\hat{r}_{\hat{f}},\hat{r})+8c_2H^2V_{\max}^2\log(|\mathcal{P}|H\delta^{-1})+16H^2V_{\max}^2\log(|\mathcal{F}|\delta^{-1})$$

$$\leq 4\hat{L}_{\mathsf{VA}}(\hat{r}_{Q^{\pi^\star}},\hat{r})+8c_2H^2V_{\max}^2\log(|\mathcal{P}|H\delta^{-1})+16H^2V_{\max}^2\log(|\mathcal{F}|\delta^{-1})$$

$$\leq 12N\mathbb{E}_{\tau^0,\tau^1\sim\mu}[(\Delta(r^\star\tau^0,\tau^1)-\Delta(\hat{r};\tau^0,\tau^1))^2]$$

$$\quad +24c_2H^2V_{\max}^2\log(|\mathcal{P}|H\delta^{-1})+80H^2V_{\max}^2\log(|\mathcal{F}|\delta^{-1})$$

where we used the fact that $r_{Q^{\pi^\star}}=r^\star$. Therefore, it holds that

$$E_{\tau^0,\tau^1\sim\mu}[(\Delta(r^\star;\tau^0,\tau^1)-\Delta(r_{\hat{f}};\tau^0,\tau^1))^2]$$

$$\leq 2E_{\tau^0,\tau^1\sim\mu}[(\Delta(r^\star;\tau^0,\tau^1)-\Delta(\hat{r};\tau^0,\tau^1))^2]+2E_{\tau^0,\tau^1\sim\mu}[(\Delta(\hat{r};\tau^0,\tau^1)-\Delta(r_{\hat{f}};\tau^0,\tau^1))^2]$$

$$\leq 26E_{\tau^0,\tau^1\sim\mu}[(\Delta(r^\star;\tau^0,\tau^1)-\Delta(\hat{r};\tau^0,\tau^1))^2]+\frac{H^2V_{\max}^2[24c_2\log(|\mathcal{P}|H\delta^{-1})+80\log(|\mathcal{F}|\delta^{-1})]}{N}.$$

Now Lemma D.4 concludes the proof. $\qquad\square$

**Lemma C.3.** *For any $f\in\mathcal{F}$, the following inequalities hold:*

$$\hat{L}_{VA}(\hat{r}_f,\hat{r})\leq 2\hat{L}_{VA}(r_f,\hat{r})+4HV_{max}^2\sum_{n=1}^{N}\sum_{h=1}^{H}\sum_{j\in\{0,1\}}\left\|P_h^\star-\hat{P}_h\right\|_1^2(s_h^{n,j},a_h^{n,j}),$$

$$-\hat{L}_{VA}(\hat{r}_f,\hat{r})\leq -\frac{1}{2}\hat{L}_{VA}(r_f,\hat{r})+2HV_{max}^2\sum_{n=1}^{N}\sum_{h=1}^{H}\sum_{j\in\{0,1\}}\left\|P_h^\star-\hat{P}_h\right\|_1^2(s_h^{n,j},a_h^{n,j}),$$

*Proof.* By definition, we have that

$$\hat{r}_f(\tau^{n,j})-r_f(\tau^{n,j})=\sum_{h=1}^{H}(P_h^\star-\hat{P}_h)V_{h+1,f}(s_h^{n,j},a_h^{n,j}).$$

Therefore, it holds that

$$\hat{L}_{\mathsf{VA}}(\hat{r}_f,\hat{r})=\sum_{n=1}^{N}\left(\hat{r}_f(\tau^{n,0})-\hat{r}_f(\tau^{n,1})-\hat{r}(\tau^{n,0})+\hat{r}(\tau^{n,1})\right)^2$$

$$=2\sum_{n=1}^{N}\left(r_f(\tau^{n,0})-r_f(\tau^{n,1})-\hat{r}(\tau^{n,0})+\hat{r}(\tau^{n,1})\right)^2$$

$$+2\sum_{n=1}^{N}\left(\sum_{h=1}^{H}\left((P_h^\star-\hat{P}_h)V_{h+1,f}(a_h^{n,0},a_h^{n,0})-(P_h^\star-\hat{P}_h)V_{h+1,f}(a_h^{n,1},a_h^{n,1})\right)\right)^2$$

$$\leq 2\hat{L}_{\mathsf{VA}}(r_f,\hat{r})+4HV_{\max}^2\sum_{n=1}^{N}\sum_{h=1}^{H}\sum_{j\in\{0,1\}}\left\|P_h^\star-\hat{P}_h\right\|_1^2(s_h^{n,j},a_h^{n,j}).$$

where we use the Cauchy-Schwarz inequality. Similarly, using $-(x+y)^2 \leq -\frac{1}{2}x^2 + y^2 \forall x, y \in \mathbb{R}$, we also have

$$-\hat{L}_{\text{VA}}(\hat{r}_f, \hat{r}) \leq -\frac{1}{2}\hat{L}_{\text{VA}}(r_f, \hat{r}) + 2HV_{\max}^2 \sum_{n=1}^{N}\sum_{h=1}^{H}\sum_{j \in \{0,1\}} \left\|P_h^\star - \hat{P}_h\right\|_1^2 (s_h^{n,j}, a_h^{n,j}).$$

$\square$

## C.2 PROOF OF THEOREM B.1

Now we prove Theorem B.1. Recall that the soft policy class $\Pi_{\mathcal{F},T}^{\text{soft}}$ is defined as

$$\Pi_{\mathcal{F},T}^{\text{soft}} = \{\pi \mid \pi_h \propto \exp(\eta \sum_{i=1}^{t} f_h^i) \ \forall h \in [H], f^1, \ldots, f^t \in \mathcal{F}, t \in [T]\}$$

for constant $\eta > 0$ whose value is specified by Lemma E.3. It is clear that $\log |\Pi_{\mathcal{F},T}^{\text{soft}}| \leq T \log |\mathcal{F}|$. Throughout the proof, we write $r^t = r_f^{\pi^t}$ and $\hat{r}^t = \hat{r}_f^{\pi^t}$ for convenience.

*Proof of Theorem B.1.* By Lemma E.1 and the fact $\hat{\pi} = \frac{1}{T}\sum_{t=1}^{T}\pi^t$, we have that

$$V_1^{\pi^\star}(s_1) - V_1^{\hat{\pi}}(s_1) = \frac{1}{T}\sum_{t=1}^{T}\left(V_1^{\pi^\star}(s_1) - V_1^{\pi^t}(s_1)\right)$$

$$= \frac{1}{T}\sum_{t=1}^{T}\left(\mathbb{E}_{\tau \sim \pi^\star}[r^t(\tau) - r^\star(\tau)] + f_1^t(s_1, \pi^t) - V_1^{\pi^t}(s_1) + V_{1,r^t}^{\pi^\star}(s_1) - V_{1,r^t}^{\pi^t}(s_1)\right)$$

$$= \underbrace{\frac{1}{T}\sum_{t=1}^{T}\mathbb{E}_{\tau^0 \sim \pi^\star, \tau^1 \sim \mu}[r^t(\tau^0) - r^\star(\tau^0) - r^t(\tau^1) + r^\star(\tau^1)]}_{\text{(I)}}$$

$$+ \underbrace{\frac{1}{T}\sum_{t=1}^{T}\left(f_1^t(s_1, \pi^t) - V_1^{\pi^t}(s_1) + \mathbb{E}_{\tau \sim \mu}[r^\star(\tau) - r^t(\tau)]\right)}_{\text{(II)}}$$

$$+ \underbrace{\frac{1}{T}\sum_{t=1}^{T}\left(V_{1,r^t}^{\pi^\star}(s_1) - V_{1,r^t}^{\pi^t}(s_1)\right)}_{\text{(III)}}$$

The term (II) is bounded by Lemma C.4 and (III) is bounded by Lemma E.3. For the term (I), note that Definition 4 implies

$$\mathbb{E}_{\tau^0 \sim \pi^\star, \tau^1 \sim \mu}[r^t(\tau^0) - r^\star(\tau^0) - r^t(\tau^1) + r^\star(\tau^1)]$$

$$= \mathbb{E}_{\tau^0 \sim \pi^\star, \tau^1 \sim \mu}[\Delta(r^t; \tau^0, \tau^1) - \Delta(r^\star; \tau^0, \tau^1)]$$

$$\leq \sqrt{\left(\mathbb{E}_{\tau^0 \sim \pi^\star, \tau^1 \sim \mu}[\Delta(r^t; \tau^0, \tau^1) - \Delta(r^\star; \tau^0, \tau^1)]\right)^2}$$

$$\leq \sqrt{\left(\frac{\left(\mathbb{E}_{\tau^0 \sim \pi^\star, \tau^1 \sim \mu}[\Delta(r^t; \tau^0, \tau^1) - \Delta(r^\star; \tau^0, \tau^1)]\right)^2}{\mathbb{E}_{\tau^0, \tau^1 \sim \mu}[(\Delta(r^t; \tau^0, \tau^1) - \Delta(r^\star; \tau^0, \tau^1))^2]}\right)\mathbb{E}_{\tau^0, \tau^1 \sim \mu}[(\Delta(r^t; \tau^0, \tau^1) - \Delta(r^\star; \tau^0, \tau^1))^2]}$$

$$\leq \sqrt{\mathcal{C}_{\mu,T}^\star(\mathcal{F})^2 \mathbb{E}_{\tau^0, \tau^1 \sim \mu}[(\Delta(r^t; \tau^0, \tau^1) - \Delta(r^\star; \tau^0, \tau^1))^2]}$$

$$\leq \frac{\alpha}{2}\mathcal{C}_{\mu,T}^\star(\mathcal{F})^2 + \frac{1}{2\alpha}\mathbb{E}_{\tau^0, \tau^1 \sim \mu}[(\Delta(r^t; \tau^0, \tau^1) - \Delta(r^\star; \tau^0, \tau^1))^2]$$

$$= \frac{\alpha}{2}\mathcal{C}_{\mu,T}^\star(\mathcal{F})^2 + \frac{1}{2\alpha N}L_{\text{VA}}(r^t, r^\star)$$

$$\leq \frac{\alpha}{2}\mathcal{C}_{\mu,T}^\star(\mathcal{F})^2 + \frac{1}{\alpha N}L_{\text{VA}}(r^t, \hat{r}) + \frac{1}{\alpha N}L_{\text{VA}}(\hat{r}, r^\star)$$

where the third inequality uses Definition 4, and we use the AM-GM inequality with constant $\alpha > 0$ in the second last step.

Combining the results, we have

$$V_1^{\pi^\star}(s_1) - V_1^{\hat\pi}(s_1)$$

$$\leq \frac{\alpha}{2}\mathcal{C}_{\mu,T}^\star(\mathcal{F})^2 + (\frac{1}{2\alpha N} - \frac{1}{4\lambda})L_{\text{VA}}(r^t, \hat r) + (\frac{1}{\alpha N} + \frac{3}{\lambda})L_{\text{VA}}(\hat r, r^\star) + V_{\max}H\sqrt{\frac{\log|\mathcal{A}|}{2T}}$$

$$+ 4V_{\max}H\sqrt{\frac{\log(|\mathcal{F}||\Pi_{\mathcal{F},T}^{\text{soft}}|\delta^{-1}) + c_2\log(|\mathcal{P}|H\delta^{-1})}{N}}$$

$$+ \frac{16}{\lambda}\left(c_2 H^2 V_{\max}^2\log(|\mathcal{P}|H\delta^{-1}) + H^2 V_{\max}^2\log(|\mathcal{F}||\Pi_{\mathcal{F},T}^{\text{soft}}|\delta^{-1}) + c_1\kappa(N/M)\log(|\mathcal{R}|\delta^{-1})\right)$$

$$\leq \frac{\alpha}{2}\mathcal{C}_{\mu,T}^\star(\mathcal{F})^2 + (\frac{1}{2\alpha N} - \frac{1}{4\lambda})L_{\text{VA}}(r^t, \hat r) + (\frac{1}{\alpha N} + \frac{3}{\lambda})\frac{c_1 N\kappa^2\log(|\mathcal{R}|^2\delta^{-1})}{M} + V_{\max}H\sqrt{\frac{\log|\mathcal{A}|}{2T}}$$

$$+ 4V_{\max}H\sqrt{\frac{\log(|\mathcal{F}||\Pi_{\mathcal{F},T}^{\text{soft}}|\delta^{-1}) + c_2\log(|\mathcal{P}|H\delta^{-1})}{N}}$$

$$+ \frac{16}{\lambda}\left(c_2 H^2 V_{\max}^2\log(|\mathcal{P}|H\delta^{-1}) + H^2 V_{\max}^2\log(|\mathcal{F}||\Pi_{\mathcal{F},T}^{\text{soft}}|\delta^{-1}) + c_1\kappa(N/M)\log(|\mathcal{R}|\delta^{-1})\right)$$

Setting $\lambda = \alpha N/2$ and $\alpha = \sqrt{\frac{c_2 H^2 V_{\max}^2\log(|\mathcal{P}|H\delta^{-1}) + H^2 V_{\max}^2\log(|\mathcal{F}||\Pi_{\mathcal{F},T}^{\text{soft}}|\delta^{-1}) + c_1\kappa(N/M)\log(|\mathcal{R}|\delta^{-1})}{N\mathcal{C}_{\mu,T}^\star(\mathcal{F})^2}}$ and using the fact $\log|\Pi_{\mathcal{F},T}^{\text{soft}}| \leq T\log|\mathcal{F}|$, we have

$$V_1^{\pi^\star} - V_1^{\hat\pi}$$

$$\leq \mathcal{O}\left(\sqrt{\mathcal{C}_{\mu,T}^\star(\mathcal{F})^2\left(\frac{V_{\max}^2 H^2(\log(|\mathcal{P}|H\delta^{-1}) + T\log(|\mathcal{F}|\delta^{-1}))}{N} + \frac{\kappa^2\log(|\mathcal{R}|\delta^{-1}))}{M}\right)} + V_{\max}H\sqrt{\frac{\log|\mathcal{A}|}{2T}}\right).$$

This concludes the proof. $\qquad\square$

**Lemma C.4.** *With probability at least $1 - 4\delta$, we have*

$$f_1^t(s_1, \pi^t) - V_1^{\pi^t}(s_1) - \mathbb{E}_{\tau\sim\mu}[r^\star(\tau) - r^t(\tau)] - \frac{1}{\lambda}\left(3L_{VA}(r^\star, \hat r) - \frac{1}{4}L_{VA}(r^t, \hat r)\right)$$

$$\leq 4V_{max}H\sqrt{\frac{\log(|\mathcal{F}||\Pi_{\mathcal{F},T}^{soft}|\delta^{-1}) + c_2\log(|\mathcal{P}|H\delta^{-1})}{N}}$$

$$+ \frac{16}{\lambda}\left(c_2 H^2 V_{max}^2\log(|\mathcal{P}|H\delta^{-1}) + H^2 V_{max}^2\log(|\mathcal{F}||\Pi_{\mathcal{F},T}^{soft}|\delta^{-1}) + c_1\kappa(N/M)\log(|\mathcal{R}|\delta^{-1})\right)$$

*for all $t \in [T]$, where $c_1, c_2$ are some absolute constants.*

*Proof.* The optimality of $f^t$ (Line 4 in Algorithm 2) implies that

$$\lambda\left(f_1^t(s_1, \pi^t) - \frac{1}{N}\sum_{n=1}^N \hat r^t(\tau^{n,0})\right) + \hat L_{\text{VA}}(\hat r^t, \hat r) \leq \lambda\left(Q_1^{\pi^t}(s_1, \pi^t) - \frac{1}{N}\sum_{n=1}^N \hat r_{Q^{\pi^t}}^{\pi^t}(\tau^n)\right) + \hat L_{\text{VA}}(\hat r_{Q^{\pi^t}}^{\pi^t}, \hat r)$$

$$= \lambda\left(V_1^{\pi^t}(s_1) - \frac{1}{N}\sum_{n=1}^N \hat r_{Q^{\pi^t}}^{\pi^t}(\tau^n)\right) + \hat L_{\text{VA}}(\hat r_{Q^{\pi^t}}^{\pi^t}, \hat r).$$

Combining this with Lemma C.3 and Lemma D.5, we have

$$f_1^t(s_1, \pi^t) - V_1^{\pi^t}(s_1)$$

$$\leq \frac{1}{N}\sum_{n=1}^{N}\hat{r}^t(\tau^{n,0}) - \frac{1}{N}\sum_{n=1}^{N}\hat{r}_{Q^{\pi^t}}^{\pi^t}(\tau^{n,0}) + \frac{1}{\lambda}\left(\hat{L}_{\mathrm{VA}}(\hat{r}_{Q^{\pi^t}}^{\pi^t}, \hat{r}) - \hat{L}_{\mathrm{VA}}(\hat{r}^t, \hat{r})\right)$$

$$\leq \frac{1}{N}\sum_{n=1}^{N}r^t(\tau^{n,0}) - \frac{1}{N}\sum_{n=1}^{N}r_{Q^{\pi^t}}^{\pi^t}(\tau^{n,0}) + \frac{1}{\lambda}\left(2\hat{L}_{\mathrm{VA}}(r_{Q^{\pi^t}}^{\pi^t}, \hat{r}) - \frac{1}{2}\hat{L}_{\mathrm{VA}}(r^t, \hat{r})\right)$$

$$+ \frac{V_{\max}}{N}\sum_{n=1}^{N}\sum_{h=1}^{H}\left\|P_h^\star - \hat{P}_h\right\|_1 (s_h^{n,0}, a_h^{n,0}) + \frac{8HV_{\max}^2}{\lambda}\sum_{n=1}^{N}\sum_{h=1}^{H}\sum_{j\in\{0,1\}}\left\|P_h^\star - \hat{P}_h\right\|_1^2 (s_h^{n,j}, a_h^{n,j})$$

$$\leq \frac{1}{N}\sum_{n=1}^{N}r^t(\tau^{n,0}) - \frac{1}{N}\sum_{n=1}^{N}r_{Q^{\pi^t}}^{\pi^t}(\tau^{n,0}) + \frac{1}{\lambda}\left(2\hat{L}_{\mathrm{VA}}(r_{Q^{\pi^t}}^{\pi^t}, \hat{r}) - \frac{1}{2}\hat{L}_{\mathrm{VA}}(r^t, \hat{r})\right)$$

$$+ V_{\max}H\sqrt{\frac{c_2\log(|\mathcal{P}|H\delta^{-1})}{N}} + \frac{8c_2 H^2 V_{\max}^2 \log(|\mathcal{P}|H\delta^{-1})}{\lambda}.$$

Note that $r_{Q^{\pi^t}}^{\pi^t} = r^\star$ by definition. Using the concentration inequalities in Lemma D.2 and Lemma D.4, we further have

$$f_1^t(s_1, \pi^t) - V_1^{\pi^t}(s_1)$$

$$\leq \frac{1}{N}\sum_{n=1}^{N}r^t(\tau^{n,0}) - \frac{1}{N}\sum_{n=1}^{N}r_{Q^{\pi^t}}^{\pi^t}(\tau^{n,0}) + \frac{1}{\lambda}\left(2\hat{L}_{\mathrm{VA}}(r_{Q^{\pi^t}}^{\pi^t}, \hat{r}) - \frac{1}{2}\hat{L}_{\mathrm{VA}}(r^t, \hat{r})\right)$$

$$+ V_{\max}H\sqrt{\frac{c_2\log(|\mathcal{P}|H\delta^{-1})}{N}} + \frac{8c_2 H^2 V_{\max}^2 \log(|\mathcal{P}|H\delta^{-1})}{\lambda}$$

$$\leq \mathbb{E}_{\tau\sim\mu}[r^t(\tau) - r^\star(\tau)] + \frac{1}{\lambda}\left(3L_{\mathrm{VA}}(r^\star, \hat{r}) - \frac{1}{4}L_{\mathrm{VA}}(r^t, \hat{r})\right) + V_{\max}H\sqrt{\frac{c_2\log(|\mathcal{P}|H\delta^{-1})}{N}}$$

$$+ \frac{1}{\lambda}(8c_2 H^2 V_{\max}^2 \log(|\mathcal{P}|H\delta^{-1}) + 16H^2 V_{\max}^2 \log(|\mathcal{F}||\Pi_{\mathcal{F},T}^{\mathrm{soft}}|\delta^{-1})) + 2HV_{\max}\sqrt{\frac{2\log(|\mathcal{F}||\Pi_{\mathcal{F},T}^{\mathrm{soft}}|\delta^{-1})}{N}}.$$

This is the desired result. $\square$

**Lemma C.5.** *For any $f\in\mathcal{F}$ and $\pi\in\Pi_{\mathcal{F},T}^{soft}$, the following inequalities hold:*

$$\hat{L}_{VA}(\hat{r}_f^\pi, \hat{r}) \leq 2\hat{L}_{VA}(r_f^\pi, \hat{r}) + 4HV_{max}^2\sum_{n=1}^{N}\sum_{h=1}^{H}\sum_{j\in\{0,1\}}\left\|P_h^\star - \hat{P}_h\right\|_1^2 (s_h^{n,j}, a_h^{n,j}),$$

$$-\hat{L}_{VA}(\hat{r}_f^\pi, \hat{r}) \leq -\frac{1}{2}\hat{L}_{VA}(r_f^\pi, \hat{r}) + 2HV_{max}^2\sum_{n=1}^{N}\sum_{h=1}^{H}\sum_{j\in\{0,1\}}\left\|P_h^\star - \hat{P}_h\right\|_1^2 (s_h^{n,j}, a_h^{n,j}),$$

*and*

$$\left|\frac{1}{N}\sum_{n=1}^{N}\hat{r}_f^\pi(\tau^{n,0}) - \frac{1}{N}\sum_{n=1}^{N}r_f^\pi(\tau^{n,0})\right| \leq \frac{V_{max}}{N}\sum_{n=1}^{N}\sum_{h=1}^{H}\left\|P_h^\star - \hat{P}_h\right\|_1 (s_h^{n,0}, a_h^{n,0}).$$

*Proof.* The proofs for the first and the second inequalities are almost identical to that of Lamma C.3. The third inequality is obtained by

$$\left|\frac{1}{N}\sum_{n=1}^{N}\hat{r}_f^\pi(\tau^{n,0}) - \frac{1}{N}\sum_{n=1}^{N}r_f^\pi(\tau^{n,0})\right| = \left|\frac{1}{N}\sum_{n=1}^{N}\sum_{h=1}^{H}(P_h^\star - \hat{P}_h)(f_{h+1}^\pi)(s_h^{n,0}, a_h^{n,0})\right|$$

$$\leq \frac{V_{\max}}{N}\sum_{n=1}^{N}\sum_{h=1}^{H}\left\|P_h^\star - \hat{P}_h\right\|_1 (s_h^{n,0}, a_h^{n,0}).$$

where we use the notation $f_h^\pi : \mathcal{S} \to [-V_{\max}, V_{\max}]$ to denote the function satisfying $f_h^\pi(s) = \mathbb{E}_{a\sim\pi_h}[f(s,a)]$ for all $s\in\mathcal{S}$. $\square$

# D CONCENTRATION LEMMAS

**Lemma D.1.** *Given $\hat{r}$, with probability at least $1 - \delta$, for all $f \in \mathcal{F}$ and $\pi \in \Pi_{\mathcal{F},T}^{soft}$, it holds that*

$$\left| \frac{1}{N} \sum_{n=1}^{N} r_f^{\pi}(\tau^{n,0}) - \mathbb{E}_{\tau \sim \mu}[r_f^{\pi}(\tau)] \right| \leq 2HV_{max}\sqrt{\frac{2\log(|\mathcal{F}||\Pi_{\mathcal{F},T}^{soft}|\delta^{-1})}{N}}$$

*Proof.* Fix $f \in \mathcal{F}$. Azuma-Hoeffding inequality implies, with probability at least $1 - \delta$,

$$\left| \frac{1}{N} \sum_{n=1}^{N} r_f^{\pi}(\tau^{n,0}) - \mathbb{E}_{\tau \sim \mu}[r_f^{\pi}(\tau)] \right| \leq 2HV_{\max}\sqrt{\frac{2\log(\delta^{-1})}{N}}.$$

The union bound over all $f \in \mathcal{F}$ and $\pi \in \Pi_{\mathcal{F},T}^{\text{soft}}$ concludes the proof. $\qquad\square$

**Lemma D.2.** *Given $\hat{r}$, with probability at least $1 - 2\delta$, for all $f \in \mathcal{F}$, it holds that*

$$L_{VA}(r_f, \hat{r}) \leq 2\hat{L}_{VA}(r_f, \hat{r}) + 16H^2V_{max}^2 \log(|\mathcal{F}|\delta^{-1})$$

*and*

$$\hat{L}_{VA}(r_f, \hat{r}) \leq \frac{3}{2}L_{VA}(r_f, \hat{r}) + 8H^2V_{max}^2 \log(|\mathcal{F}|\delta^{-1})$$

*Proof.* Fix $f \in \mathcal{F}$, then define filtration $\mathfrak{F}_n = \sigma(\tau^{1,0}, \tau^{1,1}, \ldots, \tau^{n,0}, \tau^{n,1})$ (we will use $\mathbb{E}_n[\cdot]$ to denote $\mathbb{E}[\cdot \mid \mathfrak{F}_n]$) and

$$X_n(f) = \mathbb{E}_n[(\Delta(r_f; \tau^{n,0}, \tau^{n,1}) - \Delta(\hat{r}; \tau^{n,0}, \tau^{n,1}))^2]$$
$$- (\Delta(r_f; \tau^{n,0}, \tau^{n,1}) - \Delta(\hat{r}; \tau^{n,0}, \tau^{n,1}))^2$$

so that $X_n(f) \in \mathfrak{F}_n$. With this random process, we have $\mathbb{E}_n[X_n(f)] = 0$ and

$$\mathbb{E}_n[X_n^2(f)]$$
$$= \mathbb{E}_n[(\Delta(r_f; \tau^{n,0}, \tau^{n,1}) - \Delta(\hat{r}; \tau^{n,0}, \tau^{n,1}))^4] - \mathbb{E}_n[(\Delta(r_f; \tau^{n,0}, \tau^{n,1}) - \Delta(\hat{r}; \tau^{n,0}, \tau^{n,1}))^2]^2$$
$$\leq 4H^2V_{\max}^2 \mathbb{E}_n[(\Delta(r_f; \tau^{n,0}, \tau^{n,1}) - \Delta(\hat{r}; \tau^{n,0}, \tau^{n,1}))^2]$$

Freedman's inequality (Lemma E.5) implies that, with probability at least $1 - \delta$,

$$\sum_{n=1}^{N} X_n(f) \leq \zeta \sum_{n=1}^{N} \mathbb{E}_n[X_n^2(f)] + \frac{\log(\delta^{-1})}{\zeta}$$

$$\leq 4H^2V_{\max}^2\zeta \sum_{n=1}^{N} \mathbb{E}_n[(\Delta(r_f; \tau^{n,0}, \tau^{n,1}) - \Delta(\hat{r}; \tau^{n,0}, \tau^{n,1}))^2] + \frac{\log(\delta^{-1})}{\zeta}$$

for any $\zeta \in [0, 1/4H^2V_{\max}^2]$. Setting $\zeta = 1/8H^2V_{\max}^2$, we obtain

$$\sum_{n=1}^{N} \mathbb{E}_n[(\Delta(r_f; \tau^{n,0}, \tau^{n,1}) - \Delta(\hat{r}; \tau^{n,0}, \tau^{n,1}))^2]$$

$$\leq 2 \sum_{n=1}^{N} (\Delta(r_f; \tau^{n,0}, \tau^{n,1}) - \Delta(\hat{r}; \tau^{n,0}, \tau^{n,1}))^2 + 16H^2V_{\max}^2 \log(\delta^{-1}),$$

which is equivalent to $L_{\text{VA}}(r_f, \hat{r}) \leq 2\hat{L}_{\text{VA}}(r_f, \hat{r}) + 16H^2V_{\max}^2 \log(\delta^{-1})$. We prove the first result by taking a union bound over all $f \in \mathcal{F}$. To prove the second result, consider $-X_n(f)$ and follow the same logic with $\zeta = 1/8H^2V_{\max}^2$. $\qquad\square$

**Lemma D.3.** *Given $\hat{r}$, with probability at least $1 - 2\delta$, for all $f \in \mathcal{F}$ and $\pi \in \Pi_{\mathcal{F},T}^{soft}$, it holds that*

$$L_{VA}(r_f^{\pi}, \hat{r}) \leq 2\hat{L}_{VA}(r_f^{\pi}, \hat{r}) + 16H^2V_{max}^2 \log(|\mathcal{F}||\Pi_{\mathcal{F},T}^{soft}|\delta^{-1})$$

*and*

$$\hat{L}_{VA}(r_f^{\pi}, \hat{r}) \leq \frac{3}{2}L_{VA}(r_f^{\pi}, \hat{r}) + 8H^2V_{max}^2 \log(|\mathcal{F}||\Pi_{\mathcal{F},T}^{soft}|\delta^{-1})$$

*Proof.* The proof is almost identical to that of Lemma D.2. We further consider a union bound over $\Pi_{\mathcal{F},T}^{\text{soft}}$. □

**Lemma D.4** (Lemma 2 in Zhan et al. (2024a)). *With probability at least $1 - \delta$, we have*

$$\frac{1}{N} L_{VA}(r^\star, \hat{r}) = \mathbb{E}_{\tau^0, \tau^1 \sim \mu}[(\Delta(r^\star; \tau^0, \tau^1) - \Delta(\hat{r}; \tau^0, \tau^1))^2] \leq \frac{c_1 \kappa^2 \log(|\mathcal{R}|\delta^{-1})}{M}$$

*for some absolute constant $c_1$.*

**Lemma D.5.** *With probability $1 - \delta$, for all $k \in [K]$, $h \in [H]$, and $j \in \{0, 1\}$, it holds that*

$$\sum_{n=1}^{N} \sum_{j \in \{0,1\}} \left\| \hat{P}_h - P_h^\star \right\|_1^2 (s_h^{n,j}, a_h^{n,j}) \leq c_2 \log(|\mathcal{P}|H\delta^{-1})$$

*where $c_2$ is some absolute constant.*

*Proof.* The standard MLE guarantee (e.g. Lemma 3 in Zhan et al. (2024a)) states that, with probability at least $1 - \delta$,

$$\mathbb{E}_{(s_h, a_h) \sim \mu} \left[ \left\| \hat{P}_h - P_h^\star \right\|_1^2 (s_h, a_h) \right] \lesssim \frac{\log(|\mathcal{P}|H\delta^{-1})}{N} \tag{8}$$

for all $h \in [H]$.

For a fixed $P_h \in \mathcal{P}_h$, define $X_n(P, h) = \|P_h - P_h^\star\|_1^2 (s_h^{n,0}, a_h^{n,0})$ and $\mathfrak{F}_n = \sigma(\tau^{1,0}, \tau^{1,1}, \ldots, \tau^{n,0}, \tau^{t,1})$. Applying Lemma E.6 to $(X_n(f, h))_{n \in [N]}$, we have

$$\sum_{n=1}^{N} \|P_h - P_h^\star\|_1^2 (s_h^{n,0}, a_h^{n,0}) \leq \frac{3}{2} \sum_{n=1}^{N} \mathbb{E}_n[\|P_h - P_h^\star\|_1^2 (s_h^{n,0}, a_h^{n,0})] + 4\log(\delta^{-1})$$

$$= \frac{3}{2} N \mathbb{E}_{(s_h, a_h) \sim \mu}[\|P_h - P_h^\star\|_1^2 (s_h, a_h)] + 4\log(\delta^{-1})$$

Taking a union bound over all $P_h \in \mathcal{P}_h$ and $h \in [H]$, and repeating the same argument for $\tilde{X}_n(P, h) = \|P_h - P_h^\star\|_1^2 (s_h^{n,1}, a_h^{n,1})$, with probability at least $1 - \delta$, it holds that

$$\sum_{n=1}^{N} \sum_{j \in \{0,1\}} \|P_h - P_h^\star\|_1^2 (s_h^{n,j}, a_h^{n,j}) \leq 3N \mathbb{E}_{(s_h, a_h) \sim \mu}[\|P_h - P_h^\star\|_1^2 (s_h, a_h)] + 8\log(2|\mathcal{P}|H\delta^{-1})$$

for all $h \in [H]$ and $P \in \mathcal{P}$. Combining this with (8), we conclude the proof. □

# E  SUPPORTING LEMMAS

**Lemma E.1** (Sub-optimality Decomposition (Lemma B.4 in Nguyen-Tang & Arora (2023))). *For any $f \in \mathcal{F}$ and any policies $\pi, \tilde{\pi}$, we have*

$$V_1^\pi(s_1) - V_1^{\tilde{\pi}}(s_1) = \sum_{h=1}^{H} \mathbb{E}_\pi[\mathcal{E}_h^{\tilde{\pi}}(f)(s_h, a_h)] + f_1(s_1, \tilde{\pi}(s_1)) - V_1^{\tilde{\pi}}(s_1) + V_{1, r_f^{\tilde{\pi}}}^\pi - V_{1, r_f^{\tilde{\pi}}}^{\tilde{\pi}}$$

*where $\mathcal{E}_h^{\tilde{\pi}}(f)(s, a) = f_h(s, a) - r_h^\star(s, a) - P_h^{\star, \tilde{\pi}} f_{h+1}(s, a)$.*

**Lemma E.2** (Performance Difference Lemma). *Let $\pi, \tilde{pi}$ be any policies. For any reward $r$, we have that*

$$V_{1,r}^\pi(s_1) - V_{1,r}^{\tilde{\pi}}(s_1) = \sum_{h=1}^{H} \mathbb{E}_\pi \left[ Q_{h,r}^{\tilde{\pi}}(s_h, \pi(s_h)) - Q_{h,r}^{\tilde{\pi}}(s_h, \tilde{\pi}(s_h)) \right].$$

*Proof.* Using the identity $r_h = Q_{h,r}^\pi - P_h^\star V_{h+1,r}^\pi$, we have

$$
\begin{aligned}
V_{1,r}^\pi(s_1) - V_{1,r}^{\tilde\pi}(s_1) &= \mathbb{E}_\pi[r(s_1, a_1) + P_h^\star V_{1,r}^\pi(s_1, a_1) - Q_{1,r}^{\tilde\pi}(s_1, \tilde\pi(s_1))] \\
&= \mathbb{E}_\pi[Q_{1,r}^{\tilde\pi}(s_1, a_1) - Q_{1,r}^{\tilde\pi}(s_1, \tilde\pi(s_1)) + P_h^\star V_{2,r}^\pi(s_1, a_1) - P_h^\star V_{2,r}^{\tilde\pi}(s_1, a_1)] \\
&= \mathbb{E}_\pi[Q_{1,r}^{\tilde\pi}(s_1, a_1) - Q_{1,r}^{\tilde\pi}(s_1, \tilde\pi(s_1))] + \mathbb{E}_\pi[V_{1,r}^\pi(s_2) - V_{1,r}^{\tilde\pi}(s_2)] \\
&= \dots \\
&= \sum_{h=1}^H \mathbb{E}_\pi[Q_{h,r}^{\tilde\pi}(s_h, \pi(s_h)) - Q_{h,r}^{\tilde\pi}(s_h, \tilde\pi(s_h))].
\end{aligned}
$$

$\square$

**Lemma E.3** (Online Regret Bound (e.g. Lemma D.3 in Kang & Oh (2025))). *For any sequence of functions $\{f^t\}_{t=1}^T \in \mathcal{F}^T$, the policy update (Line 5) in Algorithm 2 with $\eta = \sqrt{\frac{\log |\mathcal{A}|}{V_{max}^2 T}}$ guarantees that*

$$
\frac{1}{T} \sum_{t=1}^T \left( V_{1,r^t}^{\pi^\star}(s_1) - V_{1,r^t}^{\pi^t}(s_1) \right) \le V_{max} H \sqrt{\frac{\log |\mathcal{A}|}{2T}}.
$$

**Lemma E.4** (Azuma-Hoeffding inequality). *Let $(X_t)_{t \le T}$ be a sequence of random variables adapted to a filtration $(\mathfrak{F}_t)_{t \le T}$. If $|X_t| \le B$ for some $B > 0$ almost surely, with probability at least $1 - \delta$, we have*

$$
\left| \sum_{t=1}^T X_t - \mathbb{E}[X_t \mid \mathfrak{F}_t] \right| \le B\sqrt{2T \log(\delta^{-1})}.
$$

**Lemma E.5** (Freedman's inequality). *Let $(X_t)_{t \le T}$ be a sequence of random variables adapted to a filtration $(\mathfrak{F}_t)_{t \le T}$. Assume $|X_t| \le B$ for some $B > 0$ and $\mathbb{E}[X_t \mid \mathfrak{F}_t] = 0$. With probability at least $1 - \delta$, for any $\eta \in [0, 1/B]$, it holds that*

$$
\sum_{t=1}^T X_t \le \eta \sum_{t=1}^T \mathbb{E}[X_t^2 \mid \mathfrak{F}_t] + \frac{\log(\delta^{-1})}{\eta}.
$$

**Lemma E.6** (Lemma 2 in Zhu & Nowak (2022)). *Let $(X_t)_{t \le T}$ be a sequence of positive random variables adapted to a filtration $(\mathfrak{F}_t)_{t \le T}$. If $X_t \le B$ almost surely for all $t$, then with probability at least $1 - \delta$, the following holds:*

$$
\sum_{t=1}^T X_t \le \frac{3}{2} \sum_{t=1}^T \mathbb{E}[X_t \mid \mathfrak{F}_t] + 4B \log(\delta^{-1}),
$$

$$
\sum_{t=1}^T \mathbb{E}[X_t \mid \mathfrak{F}_t] \le 2 \sum_{t=1}^T X_t + 8B \log(\delta^{-1})
$$

# F    ADDITIONAL EXPERIMENTS

## F.1    ON THE TRANSITION MODEL

| Dataset (# of feedback) | medium-replay (500) | medium-replay (1000) | medium-expert (500) | medium-expert (1000) |
|---|---|---|---|---|
| PVO | 66.20 | 69.14 | 68.06 | 73.00 |
| PVO with transition model | 64.08 | 71.18 | 68.46 | 71.16 |
| IQL | 53.86 | 58.76 | 57.78 | 67.84 |

Table 1: Success rates on Meta-World datasets, averaged over five random seeds. PVO and its variant with a learned transition model achieve similar performance.

The transition model in PVO plays a theoretical role in defining the value alignment loss and deriving the sample-complexity analysis, where the expectation $\mathbb{E}_{s' \sim \hat{P}(s,a)}[V(s')]$ is taken over the estimated transition model. However, in practice, this term can be efficiently approximated using transition samples from the dataset. Our practical implementation therefore follows this sample-based approximation, a design choice originating from APPO Kang & Oh (2025), which employs a variant of our value alignment loss (as discussed in Section 3.3) and similarly replaces the model expectation with dataset samples. This simplification reduces computational cost while maintaining performance.

To verify whether this simplification affects performance, we implemented a variant that explicitly trains a transition model and computes the expectation term via sampling from it. Table 1 compares this variant with the original implementation on Meta-World datasets used in Section 5, trained with 500 and 1000 preference feedbacks. The results show that both implementations achieve highly comparable performance, with neither consistently outperforming the other. These findings indicate that while the transition model is conceptually important for theoretical formulation, it is not practically necessary for effective learning. Our sample-based implementation thus offers a simpler and more computationally efficient realization of PVO.

## F.2    EMPIRICAL COMPARISON WITH FLOW TO BETTER ZHANG ET AL. (2024)

| Dataset | box-close | dial-turn | drawer-open | lever-pull | sweep-into | sweep | button-press-topdown | button-press-topdown-wall |
|---|---|---|---|---|---|---|---|---|
| PVO | $58.72_{\pm 16.61}$ | $84.32_{\pm 4.71}$ | $100.00_{\pm 0.00}$ | $96.64_{\pm 2.84}$ | $27.84_{\pm 4.36}$ | $94.08_{\pm 6.31}$ | $43.36_{\pm 19.03}$ | $48.16_{\pm 14.54}$ |
| FTB | $0.00_{\pm 0.00}$ | $0.00_{\pm 0.00}$ | $97.60_{\pm 3.39}$ | $0.00_{\pm 0.00}$ | $62.67_{\pm 23.71}$ | $0.00_{\pm 0.00}$ | $0.00_{\pm 0.00}$ | $0.00_{\pm 0.00}$ |

Table 2: Success rates on Meta-World medium-replay datasets with 1000 preference feedback. For FTB, we report the results averaged over three random seeds.

We also evaluated Flow to Better (FTB) Zhang et al. (2024) on the Meta-World medium-replay datasets with 1000 preference feedback. We used the official implementation of FTB and default hyperparameters, only adapting the episode length to be consistent with our datasets.

Interestingly, FTB performs very well on drawer-open and even outperforms PVO on sweep-into, but collapses to 0% success on all other tasks. We note that a 0% success rate does not necessarily mean that the method failed to learn at all – we do observe nontrivial improvements in episodic returns during training – but rather that the learned policies rarely satisfy the success criteria on these tasks.

Overall, these results suggest that FTB can be effective on certain tasks, yet exhibits substantial sensitivity to the dynamics and data distribution. One possible explanation is that FTB ultimately relies on imitation of generated trajectories: when the dataset contains sufficiently good trajectories, the filtering-and-cloning procedure may succeed, whereas in tasks where high-quality demonstrations are sparse, the model may end up cloning suboptimal behaviors. In contrast, PVO achieves consistently strong performance across all tasks, including those where FTB collapses, which aligns with our main goal: providing a simple, value-based offline PbRL algorithm that is both theoretically grounded and practically robust.

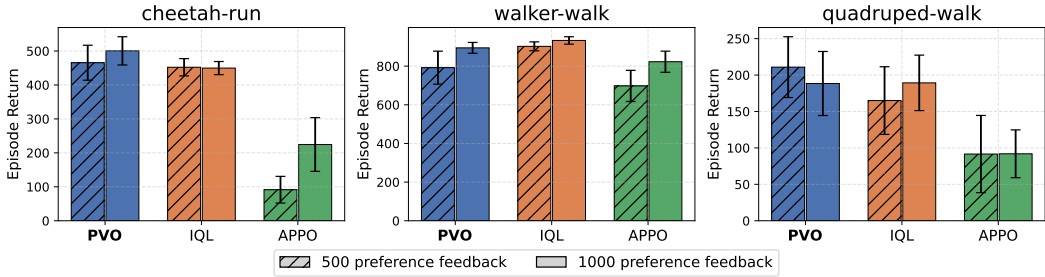

Figure 4: Performance on DMControl datasets measured by episode return. Each plot displays the mean and standard deviation over five random seeds.

## F.3 EVALUATION ON DMCONTROL DATASETS

We present additional experimental results that were omitted from Section 5 due to space constraints. Figure 4 shows the performance evaluation on the DMControl Tassa et al. (2018) dataset. Overall, PVO outperforms APPO by a large margin, which is consistent with the results in Section 5. Compared to IQL, PVO performs better in the cheetah-run and quadruped-walk datasets, while showing comparable performance in the walker-walk dataset.

## F.4 COMPLETE NUMERICAL RESULTS

We provide the complete numerical results in the tables below. The results better than 95% of the best performance are highlighted.

| Dataset and # of feedback | PT | DPPO | IPL | IQL | APPO | PVO (ours) |
|---|---|---|---|---|---|---|
| box-close-500 | $0.33 \pm 1.16$ | $10.20 \pm 11.47$ | $5.93 \pm 5.81$ | $27.84 \pm 28.52$ | $18.24 \pm 15.60$ | $\mathbf{31.04} \pm 22.44$ |
| dial-turn-500 | $68.67 \pm 12.39$ | $26.67 \pm 22.23$ | $31.53 \pm 12.50$ | $74.56 \pm 10.32$ | $\mathbf{80.96} \pm 4.49$ | $\mathbf{82.72} \pm 6.49$ |
| drawer-open-500 | $88.73 \pm 11.64$ | $35.93 \pm 11.18$ | $19.00 \pm 13.63$ | $\mathbf{98.24} \pm 3.52$ | $87.68 \pm 10.04$ | $\mathbf{99.52} \pm 0.96$ |
| lever-pull-500 | $82.40 \pm 22.69$ | $10.13 \pm 12.19$ | $31.20 \pm 15.76$ | $85.28 \pm 3.18$ | $75.76 \pm 7.17$ | $\mathbf{93.92} \pm 4.92$ |
| sweep-into-500 | $20.53 \pm 8.26$ | $23.07 \pm 7.02$ | $32.20 \pm 7.35$ | $20.96 \pm 7.91$ | $24.08 \pm 5.91$ | $\mathbf{37.92} \pm 11.00$ |
| sweep-500 | $43.07 \pm 24.57$ | $10.47 \pm 15.84$ | $27.20 \pm 23.81$ | $\mathbf{90.88} \pm 6.25$ | $26.80 \pm 5.32$ | $87.68 \pm 8.97$ |
| button-press-topdown-500 | $22.87 \pm 9.06$ | $3.93 \pm 4.34$ | $34.73 \pm 13.92$ | $18.88 \pm 12.14$ | $\mathbf{53.52} \pm 13.86$ | $\mathbf{56.80} \pm 13.28$ |
| button-press-topdown-wall-500 | $0.87 \pm 1.43$ | $0.80 \pm 1.51$ | $8.93 \pm 9.84$ | $14.24 \pm 5.27$ | $\mathbf{64.32} \pm 20.99$ | $39.36 \pm 14.52$ |
| box-close-1000 | $2.27 \pm 2.86$ | $9.33 \pm 9.60$ | $6.73 \pm 8.41$ | $40.96 \pm 22.04$ | $34.24 \pm 18.49$ | $\mathbf{58.72} \pm 16.61$ |
| dial-turn-1000 | $68.60 \pm 5.50$ | $36.40 \pm 21.95$ | $43.93 \pm 13.37$ | $77.28 \pm 10.64$ | $81.44 \pm 6.73$ | $\mathbf{84.32} \pm 4.71$ |
| drawer-open-1000 | $\mathbf{95.40} \pm 7.27$ | $36.47 \pm 7.30$ | $28.53 \pm 18.37$ | $\mathbf{99.52} \pm 0.64$ | $\mathbf{98.56} \pm 2.68$ | $\mathbf{100.00} \pm 0.00$ |
| lever-pull-1000 | $72.93 \pm 10.16$ | $8.53 \pm 9.96$ | $40.40 \pm 17.38$ | $87.04 \pm 5.64$ | $76.96 \pm 4.40$ | $\mathbf{96.64} \pm 2.84$ |
| sweep-into-1000 | $20.27 \pm 7.84$ | $23.33 \pm 7.80$ | $30.40 \pm 7.74$ | $24.00 \pm 5.97$ | $18.16 \pm 11.14$ | $\mathbf{27.84} \pm 4.36$ |
| sweep-1000 | $29.13 \pm 14.55$ | $8.73 \pm 16.37$ | $38.33 \pm 24.87$ | $\mathbf{98.80} \pm 1.01$ | $17.36 \pm 12.44$ | $94.08 \pm 6.31$ |
| button-press-topdown-1000 | $18.27 \pm 10.62$ | $3.20 \pm 3.04$ | $36.67 \pm 17.40$ | $20.00 \pm 5.41$ | $\mathbf{59.04} \pm 18.97$ | $43.36 \pm 19.03$ |
| button-press-topdown-wall-1000 | $2.13 \pm 2.96$ | $0.27 \pm 0.85$ | $14.07 \pm 11.47$ | $22.48 \pm 5.28$ | $\mathbf{62.96} \pm 18.38$ | $48.16 \pm 14.54$ |
| Average | 39.78 | 15.47 | 26.86 | 56.31 | 55.01 | **67.63** |

Table 3: Success rates on Meta-World `medium-replay` dataset with $500$ and $1000$ preference feedback, averaged over five random seeds. The results of PT, DPPO, and IPL are from Choi et al. (2024).

| Dataset and # of feedback | IQL | APPO | PVO (ours) |
|---|---|---|---|
| box-close-500 | 19.68 ± 11.96 | 1.92 ± 5.61 | **49.44** ± 12.19 |
| dial-turn-500 | 13.60 ± 8.95 | 42.56 ± 23.27 | **73.44** ± 22.96 |
| drawer-open-500 | **100.00** ± 0.00 | 88.48 ± 9.65 | **100.00** ± 0.00 |
| lever-pull-500 | **71.28** ± 3.30 | 11.52 ± 9.89 | 66.56 ± 18.12 |
| sweep-into-500 | **89.44** ± 6.02 | 49.60 ± 21.87 | 85.12 ± 14.86 |
| sweep-500 | **99.12** ± 0.78 | 3.52 ± 3.46 | **96.80** ± 2.48 |
| button-press-topdown-500 | 59.84 ± 18.13 | 3.20 ± 6.40 | **70.40** ± 14.20 |
| button-press-topdown-wall-500 | **9.28** ± 3.02 | 3.36 ± 6.72 | 2.72 ± 2.41 |
| box-close-1000 | 57.28 ± 7.72 | 1.92 ± 1.09 | **63.36** ± 23.67 |
| dial-turn-1000 | 26.72 ± 24.13 | 42.56 ± 12.81 | **76.64** ± 12.72 |
| drawer-open-1000 | **100.00** ± 0.00 | 80.16 ± 5.29 | **96.00** ± 8.00 |
| lever-pull-1000 | **77.68** ± 6.99 | 10.00 ± 10.18 | 69.28 ± 5.00 |
| sweep-into-1000 | **95.04** ± 1.92 | 32.96 ± 9.74 | 93.28 ± 8.34 |
| sweep-1000 | **99.28** ± 0.73 | 0.40 ± 0.80 | **98.88** ± 1.39 |
| button-press-topdown-1000 | **78.24** ± 11.15 | 14.40 ± 7.90 | **81.12** ± 5.02 |
| button-press-topdown-wall-1000 | **8.48** ± 3.94 | 0.16 ± 0.32 | **5.44** ± 7.76 |
| Average | 62.81 | 24.17 | **70.53** |

Table 4: Success rates on Meta-World `medium-expert` dataset with 500 and 1000 preference feedback, averaged over five random seeds.

| Dataset and # of feedback | IQL | APPO | PVO (ours) |
|---|---|---|---|
| cheetah-run-500 | 299.59 ± 47.96 | 91.48 ± 39.26 | **465.46** ± 51.50 |
| walker-walk-500 | **927.03** ± 19.15 | 697.69 ± 80.59 | 792.14 ± 85.36 |
| quadruped-walk-500 | **213.30** ± 48.79 | 91.58 ± 52.99 | **210.90** ± 41.72 |
| cheetah-run-1000 | 357.88 ± 17.46 | 224.60 ± 79.16 | **500.48** ± 41.60 |
| walker-walk-1000 | **931.48** ± 17.11 | 822.87 ± 54.64 | **894.65** ± 28.04 |
| quadruped-walk-1000 | **176.16** ± 78.54 | 91.92 ± 32.83 | **188.42** ± 43.95 |
| Average | **484.24** | 336.69 | **508.67** |

Table 5: Episode returns on DMControl dataset with 500 and 1000 preference feedback, averaged over five random seeds.

## G EXPERIMENTAL DETAILS

### G.1 DATASETS

| Dataset | box-close | dial-turn | sweep | sweep-into | drawer-open | lever-pull | button-press-topdown | button-press-topdown-wall |
|---|---|---|---|---|---|---|---|---|
| medium-replay | 2.4M | 900k | 2.1M | 300k | 300k | 900k | 300k | 450k |
| medium-expert | 900k | 300k | 900k | 300k | 300k | 300k | 300k | 300k |

Table 6: The sizes of Meta-World `medium-replay` datasets (Choi et al., 2024) and `medium-expert` datasets.

The DMControl `medium-replay` and Meta-World `medium-replay` datasets are created by Choi et al. (2024). The datasets are generated from the replay buffers of online SAC (Haarnoja et al., 2018) agents. Following Choi et al. (2024), we use different dataset sizes for each Meta-World task, as shown in Table 6. All DMcontrol datasets contain 300k transition samples.

We created the Meta-World `medium-expert` data using the code provided by Hejna & Sadigh (2024). For dial-turn, sweep-into, drawer-open, and lever-pull tasks, we collected 50 trajectories with an expert policy, 50 trajectories with expert policies for randomized variants and goals of the task, 100 trajectories with expert policies for different tasks, 200 trajectories with a random policy, and 200 trajectories with an $\varepsilon$-greedy expert policy that takes greedy actions with a 50% probability. In total, there are 600 trajectories (300k transitions) for each task dataset. Additionally, standard Gaussian noise was added to the actions of each policy. The dataset sizes match those of the medium-

replay dataset. For box-close and sweep tasks, we collected 1800 trajectories, maintaining the ratio of sources.

To create the mixture datasets in the ablation study, we collected 600 trajectories from an expert policy trained on multiple tasks and goals, and another 600 trajectories from a random policy. Then, for each mixture ratio $r \in \{0, 0.25, 0.5, 0.75, 1\}$, we formed a dataset consisting of $600(1 - r)$ expert trajectories and $600r$ random trajectories. For example, the $r = 0.25$ contains 450 expert trajectories and 150 random trajectories.

### G.2 IMPLEMENTATION DETAILS.

We used the official implementation of Choi et al. (2024) and Kang et al. (2023) for reward models, IQL agents, and APPO agents. The reward model is an ensemble of three fully connected neural networks with three hidden layers of 128 neurons. The Q, V, and policy are parameterized as fully connected neural networks with three hidden layers of 256 neurons. We set the hyperparameters of IQL and APPO as suggested in Choi et al. (2024) and Kang & Oh (2025), except the advantage weight of IQL which we searched over $\beta \in \{3.0, 10.0\}$. The detailed hyperparameters are listed in Table 7.

We run experiments on an Intel Xeon Gold 6226R CPU and an Nvidia GeForce RTX 3090 GPUs. For PVO, the use of a trajectory does not significantly slow the training. All algorithms take approximately 2-3 hours to complete 250k gradient steps. The performance is measured by

| Algorithm | Component | Value |
|---|---|---|
| Reward model | Neural networks | 3-layers, hidden dimension 128 |
| | Activation | ReLU for hidden layers, Tanh for final output |
| | Optimizer | Adam (Kingma & Ba, 2015) with learning rate 1e-3 |
| | Batch size | 512 |
| | Epochs | 300 |
| | Number of ensembles | 3 |
| PVO | Neural networks (Q, V, $\pi$) | 3-layers, hidden dimension 256 |
| | Activaton | ReLU for hidden layers |
| | Q,V, $\pi$ optimizer | Adam with learning rate 3e-4 |
| | Batch size | 256 |
| | Target Q soft update | 0.005 |
| | $\beta$ (IQL advantage weight) | 10.0 |
| | $\tau$ (IQL expectile parameter) | 0.7 |
| | discount factor | 0.99 |
| IQL | Neural networks (Q, V, $\pi$) | 3-layers, hidden dimension 256 |
| | Activaton | ReLU for hidden layers |
| | Q, V, $\pi$ optimizer | Adam with learning rate 3e-4 |
| | Batch size | 256 |
| | Target Q soft update | 0.005 |
| | $\beta$ (IQL advantage weight) | 10.0 |
| | $\tau$ (IQL expectile parameter) | 0.7 |
| | discount factor | 0.99 |
| APPO | Neural networks (Q, V, $\pi$) | 3-layers, hidden dimension 256 |
| | Activaton | LeakyReLU for hidden layers |
| | Q,V, $\alpha$ optimizer | Adam with learning rate 3e-4 |
| | $\pi$ optimizer | Adam with learning rate 3e-5 |
| | Batch size | 256 transitions and 16 trajectory pairs |
| | Target Q soft update | 0.001 |
| | discount factor | 0.99 |
| TD3+BC | Neural networks (Q, V, $\pi$) | 3-layers, hidden dimension 256 |
| | Activaton | ReLU for hidden layers |
| | Q, V, $\pi$ optimizer | Adam with learning rate 3e-4 |
| | Batch size | 256 |
| | Target Q soft update | 0.005 |
| | $\alpha$ (BC weight) | 2.5 |
| | Policy noise | 0.2 |
| | Policy noise clip | 0.5 |
| | discount factor | 0.99 |
| XQL | Neural networks (Q, V, $\pi$) | 3-layers, hidden dimension 256 |
| | Activaton | ReLU for hidden layers |
| | Q, V, $\pi$ optimizer | Adam with learning rate 3e-4 |
| | Batch size | 256 |
| | Target Q soft update | 0.005 |
| | $\beta$ (Gumbel regression temperature) | 1.0 |
| | discount factor | 0.99 |

Table 7: Implementation details and hyperparameters.

