# OpenReview forum: "Offline Preference-Based Value Optimization"
_ICLR.cc/2026/Conference — ICLR 2026 Poster_

### Official Review · Reviewer_frBc · 2025-10-28

**Soundness:** 2
**Presentation:** 2
**Contribution:** 2
**Rating:** 4
**Confidence:** 3

**Summary:**

To address the issues of high computational complexity and instability in PbRL, this paper proposes an optimization objective named the value alignment loss and validates the effectiveness of the proposed method both theoretically and experimentally.

**Strengths:**

1) Through experiments, the paper verifies that PVO outperforms other methods for most tasks.

2) The paper provides sufficient theoretical derivation and analysis, including a derivation of the algorithm's computational complexity.

**Weaknesses:**

1) Algorithmic Aspect: The algorithm is essentially consistent with IQL [1], showing limited innovation. IQL minimizes $(R+\gamma V-Q)^2$, while PVQ is equivalent to minimizing $(\sum_l (Q_l-\gamma V_l-R_l))^2$.

2) Theoretical Aspect: The theoretical analysis is similar to that in paper [2], offering limited contribution.

[1] Offline Reinforcement Learning with Implicit Q-Learning, ICLR, 2021.

[2] Provable offline preference-based reinforcement learning, ICLR, 2024.

**Questions:**

1) Motivation: Before Definition 1, the authors discuss learning a value function consistent with preference feedback. How is this reflected in the proposed value alignment loss?

2) IQL Parameters: The framework is similar to IQL. For a fair comparison, what is the performance of IQL when its advantage weight parameter $\beta$ is set to be consistent with PVO?

3) Transition Model: In the practical deployment described in Section 3.4, the environment model does not seem to be used. What, then, is the purpose of training the transition model?

---

> ### Author Response · Authors · 2025-11-19
>
> We thank you for your time and effort in reviewing our paper and for raising detailed questions. Your feedback has been invaluable in helping us improve the clarity and presentation of our contributions. Below, we address each question in detail.
>
> ## Q1. Motivation for the Value Alignment Loss
> Unlike standard RL, preference-based RL (PbRL) receives learning signals only from trajectory-pair feedback rather than per-step rewards. Consequently, a value function consistent with preference feedback must be learned differently from the standard TD-style approach.
>
> As discussed in Section 3.1, our motivation begins from reward-estimation error bounds. In standard RL, the reward function is known or we have an estimator $\hat{r}$ with bounded **pointwise** error
>
> $\mathbb{E}_{s,a \sim \mu}[(r^{\star}(s,a) - \hat{r}(s,a))^2].$ (*)
>
> Using the induced reward (Definition 1), for any value function $f : \mathcal{S}\times\mathcal{A} \rightarrow \mathbb{R}$, we have
>
> $r\_f(s,a) := f(s,a) - \mathbb{E}\_{s'\sim P(s,a)}[\max\_{a'} f(s',a')]$ (subscript $h$ is omitted for simplicity).
>
> Substituting $\hat{r} \leftarrow r_f$ in (*) yields the TD loss
>
> $\mathbb{E}\_{s,a \sim \mu}[(r^{\star}(s,a) + \mathbb{E}\_{s'\sim P(s,a)}[\max\_{a'} f(s',a')] - f(s,a))^2]$,
>
> so minimizing the TD loss guarantees sample-complexity bounds in standard offline RL; this value–reward correspondence underlies many theoretical analyses [2,6,7,8].
>
> In PbRL, by contrast, the reward error bound (Lemma 2 in [3]) only controls the **relative error across trajectory pairs**, not pointwise accuracy:
>
> $\mathbb{E}_{\tau^0,\tau^1\sim\mu}[(\hat{r}(\tau^0) - \hat{r}(\tau^1) - r^{\star}(\tau^0) + r^{\star}(\tau^1))^2]$. (**)
>
> Therefore, **minimizing state-action Bellman error (TD loss) alone does not yield theoretical guarantees**. However, if we substitute the induced reward for $\hat{r}$ in (**) and minimize the resulting
> **value alignment loss** (Eq. 2), we can prove **sample-complexity bounds**. By Definition 1, the value alignment loss is equivalently the difference of cumulative Bellman errors between trajectory pairs:
>
> $\mathbb{E}\_{\tau^0, \tau^1 \sim \mu}\left[ \left( \sum^H\_{h=1} (f\_h - \hat{r}\_h - P\_h V\_{h+1,f}) (s^{n,0}\_h,a^{n,0}\_h) - \sum^H\_{h=1} (f\_h - \hat{r}\_h - P\_h V\_{h+1,f}) (s^{n,1}\_h,a^{n,1}\_h) \right)^2 \right]$,
>
> which means that the value alignment loss can be interpreted as a PbRL analogue of the TD loss.
>
> In summary, because PbRL observes pairwise, trajectory-level preference feedback rather than stepwise rewards, the TD loss used by standard RL algorithms does not align with the available concentration bounds.
> Our value alignment loss is designed directly from the PbRL reward concentration bound, with induced rewards (Definition 1) providing both (i) a principled definition and (ii) a clear interpretation as a “TD-style” objective for PbRL.
>
>
> ## Why PVO Is Fundamentally Different from IQL
>
> While PVO and IQL share certain implementation components, but the algorithms address **different problem settings** and optimize **different objectives**.
> - IQL (standard RL): Designed for per-step rewards, IQL learns Q function by minimizing the squared Bellman error $(r(s,a) + \gamma V(s') - Q(s,a))^2$ which matches the pointwise-reward regime.
> - PVO (PbRL): Designed for preference feedback without per-step rewards. As shown above, the value alignment loss measures the relative error of induced reward across trajectory pairs, which is equivalent (via Definition 1) to the difference of cumulative Bellman errors between the trajectory pair. This is the correct object controlled by PbRL concentration bounds, and it is not the IQL TD loss. Notably, the value alignment loss aggregates the Bellman errors at the trajectory level before applying the squared penalty, which captures a **fundamentally different signal than summing per-step squared Bellman errors as in TD learning**.
>
> For stability in offline learning, our practical implementation employs expectile regression. However, we can utilize alternative methods to mitigate out-of-distribution error: As shown in Section 5 and Figure 3, the value alignment objective can also be instantiated with XQL [4] (Gumbel regression) or TD3+BC [5] (behavior-regularized policy), and these non-IQL instantiations consistently improve performance over TD-based baselines. Thus, **PVO is objective-level and setting-level distinct from IQL, and not a minor variant**.

---

> ### Author Response · Authors · 2025-11-19
>
> ## Novelty of Theoretical Analysis
> Our setting is inspired by [3], and we leverage their reward concentration lemma (Lemma D.4) to prove our main theorem (Theorem 4.1). Nonetheless, the algorithmic structure and analysis pipeline are fundamentally different:
> - The FREEHAND-transition algorithm in [3] is purely model-based: It relies on a value optimization oracle given a transition model and **jointly optimizes the value, reward, and transition models**. As a result, the analysis yields bounds that depend directly on model errors (reward/transition), consistent with its **model-based** nature.
> - We obtain reward/transition models via maximum-likelihood estimation, then keep them fixed and **optimize only the value function** through the value alignment loss. Because we optimize the value (not the models), our proof requires a **distinct sub-optimality decomposition via induced rewards** (Definition 1) to translate model concentration bounds (Lemma D.4, D.5) into a value-learning sample-complexity bound. This value-centric argument and decomposition are specific to our objective and are detailed in Section C.1.
>
>
>
> ## Q2. Additional Experiments with IQL
>
> | Dataset (number of feedback) | medium-replay (500) | medium-replay (1000) | medium-expert (500) | medium-expert (1000) |
> | --- | --- | --- | --- | --- |
> | PVO | 66.20 | 69.14 | 68.06| 73.00 |
> | IQL ($\beta=3.0$) | 49.20 | 50.79 | 53.75 | 61.96 |
> | IQL ($\beta=10.0$) | 53.86 | 58.76 | 57.78 | 67.84 |
>
> As noted in Section 5.1, our experimental setup follows the datasets and protocols from [1] and [2], and we adopted the default hyperparameters for IQL from those references (e.g. $\beta=3.0$).
> In response to the reviewer’s helpful comment regarding the advantage-weight parameter, we conducted additional experiments by setting it to the same value ($\beta=10.0$) as used in PVO.
>
> The results above show that increasing the advantage weight improves IQL’s performance, narrowing the gap with PVO.
> Nevertheless, **PVO continues to outperform even the optimized IQL baseline** across diverse datasets and feedback regimes, demonstrating the robustness of its value alignment formulation.
> We appreciate the reviewer’s thoughtful suggestion, which helped us strengthen our empirical evaluation and further highlight PVO’s advantages under a fairer comparison. We have **updated all experimental results including Section 5 and Appendix F** accordingly to include this optimized baseline.

---

> ### Author Response · Authors · 2025-11-19
>
> ## Q3. On the Transition Model
>
> The transition model in PVO plays a theoretical role in defining the value alignment loss and deriving the sample-complexity analysis, where the expectation $\mathbb{E}_{s'\sim \hat{P}(s,a)}[V(s')]$ is taken over the estimated transition model.
> However, in practice this term can be efficiently approximated using transition samples from the dataset, without explicitly training or querying a model.
>
> Our practical implementation therefore follows this sample-based approximation, a **design choice originating from APPO** [2], which employs a variant of our value alignment loss (as discussed in Section 3.3) and similarly replaces the model expectation with dataset samples.
> This simplification reduces computational cost while maintaining performance and stability.
>
> To verify whether this design choice affects performance, we additionally implemented a variant that explicitly trains a transition model and computes the expectation term via sampling from it.
> The table below compares this variant with the original implementation on Meta-World datasets used in Section 5.2, trained with 500 and 1000 preference feedbacks (each averaged over five random seeds).
>
> The results show that **both implementations achieve highly comparable performance**, with neither consistently outperforming the other.
> These findings indicate that while the transition model is conceptually important for theoretical formulation, it is not practically necessary for effective learning.
> Our sample-based implementation thus offers a simpler and more computationally efficient realization of PVO.
>
> | Dataset (number of feedback) | medium-replay (500) | medium-replay (1000) | medium-expert (500) | medium-expert (1000) |
> | --- | --- | --- | --- | --- |
> | PVO | 66.20 | 69.14 | 68.06| 73.00 |
> | PVO with learned transition model | 64.08 | 71.18 | 68.46 | 71.16 |
> | IQL | 53.86 | 58.76 | 57.78 | 67.84 |
>
> To summarize, our theoretical analysis is established for the formulation that includes the MLE transition model, as this is the setting in which the value alignment loss can be rigorously analyzed and bounded. In practice, however, the expectation term can be approximated directly using dataset transitions, following the implementation strategy used in prior PbRL work [2]. Our experiments show that both implementations yield very similar performance, suggesting that the choice between a learned model and a sample-based approximation is primarily an implementation decision rather than a driver of empirical performance.
>
> We appreciate the reviewer’s insightful question, which prompted us to further clarify the theoretical and practical roles of the transition model.
> We have also updated Appendix F.1 to provide the details on this analysis and the corresponding experimental results.
>
>
> [1] Heewoong Choi, Sangwon Jung, Hongjoon Ahn, and Taesup Moon. Listwise reward estimation for offline preference-based reinforcement learning. In Forty-first International Conference on Machine Learning, 2024.
>
> [2] Hyungkyu Kang and Min-hwan Oh. Adversarial policy optimization for offline preference-based reinforcement learning. In The Thirteenth International Conference on Learning Representations, 2025.
>
> [3] Wenhao Zhan, Masatoshi Uehara, Nathan Kallus, Jason D. Lee, and Wen Sun. Provable offline preference-based reinforcement learning. In The Twelfth International Conference on Learning Representations, 2024.
>
> [4] Divyansh Garg, Joey Hejna, Matthieu Geist, and Stefano Ermon. Extreme q-learning: Maxent RL without entropy. In The Eleventh International Conference on Learning Representations, 2023.
>
> [5] Scott Fujimoto and Shixiang Shane Gu. A minimalist approach to offline reinforcement learning. Advances in neural information processing systems, 34:20132–20145, 2021.
>
> [6] Andrea Zanette, Martin J Wainwright, and Emma Brunskill. Provable benefits of actor-critic methods for offline reinforcement learning. Advances in neural information processing systems, 34:13626–13640, 2021.
>
> [7] Tengyang Xie, Ching-An Cheng, Nan Jiang, Paul Mineiro, and Alekh Agarwal. Bellman-consistent pessimism for offline reinforcement learning. Advances in neural information processing systems, 34:6683–6694, 2021.
>
> [8] Thanh Nguyen-Tang and Raman Arora. On sample-efficient offline reinforcement learning: Data diversity, posterior sampling and beyond. Advances in neural information processing systems, 36:61115–61157, 2023.

---

> > ### Comment · Reviewer_frBc · 2025-11-27
> >
> > Thank you for the authors' patient responses. I have gained a basic understanding of the three questions raised in the issue section. I also recognize the considerable effort the authors have devoted to the experiments and theoretical derivations, and I understand the motivation behind the method as well as the theoretical improvements the authors have made.
> >
> > However, since related work has already explored state-action level PbRL analysis and the integration of transition models into PbRL theoretical analysis, the overall contribution of the theory and methodology remains somewhat limited. Therefore, I have only raised the score to 6.

---

### Official Review · Reviewer_BMtm · 2025-10-29

**Soundness:** 3
**Presentation:** 2
**Contribution:** 2
**Rating:** 6
**Confidence:** 2

**Summary:**

This work addresses a critical gap in offline preference-based reinforcement learning: the tradeoff between theoretical guarantees and practical usability in existing methods. By proposing Preference-based Value Optimization, which uses a value alignment loss unifying value-based and actor-critic PbRL with rate-optimal guarantees, it delivers a unified solution that excels both in theory and experiments.

**Strengths:**

1. The work targets the well-documented tradeoff in existing offline preference-based RL methods, where theoretically rigorous approaches are often computationally intractable, and practical implementations suffer from suboptimal sample complexity or training instability.
2. One advantage of the work's insight is its shift from the standard offline PbRL paradigm (first infer a reward model, then train the value function using that reward directly) to a different viewpoint: it anchors value function learning to preference alignment via an "induced reward" derived from the value function itself, rather than treating the reward model as the sole driver of value training.

**Weaknesses:**

1. The first weakness is the paper’s inconsistent formatting of mathematical formulas, specifically, the absence of terminal punctuation in Line 318.
2. The paper’s baseline set is limited in scope. Several existing works in PbRL use generative models, like trajectory generative adversarial networks and diffusion models for preference modeling, to infer preference-aligned behavior without relying on intermediate reward models. These should also be included.

**Questions:**

See weaknesses.

---

> ### Author Response · Authors · 2025-11-20
>
> We greatly appreciate the time and effort you have dedicated to reviewing our work and providing insightful feedback. Below, we provide detailed responses to your comments.
>
> ## Formatting of Mathematical Expressions
>
> We appreciate the reviewer’s attention to detail. We have carefully reviewed the formatting of all mathematical expressions and corrected the minor inconsistency noted in Line 318 by ensuring proper punctuation and consistent presentation across the paper.
>
>
> ## Discussion on Generative Models for PbRL
>
> Among generative approaches for preference modeling, the method most closely related to ours is FTB [1], which directly trains a conditional diffusion model to synthesize preferred trajectories. Although FTB is already introduced in Section 1.1, we provide a more detailed clarification of its relation to PVO below:
>
> FTB [1] leverages a conditional diffusion model that generates a positive trajectory conditioned on a negative trajectory. Utilizing the diffusion model, it progressively generate high-quality trajectories, followed by behavior cloning on the generated data. In contrast, PVO directly learns a value function aligned with preference feedback by minimizing the value alignment loss, and extracts the corresponding policy. Also, training a diffusion model for FTB requires approximately 15 hours as reported in [1], while our method takes less than 3 hours when trained on a GTX 3090 GPU.
> The relative advantages are as follows: PVO offers a sample complexity guarantee and is computationally more efficient overall, whereas FTB offers flexibility in handling non-Markovian preference and conducts a broader set of empirical evaluations.
>
> Another generative PbRL approach, OPPO [2], is likewise introduced in Section 1.1. OPPO applies hindsight information matching (HIM) [3] to preference-based RL. Through HIM, it trains an encoder that maps trajectories into a latent space and a policy conditioned on the latent variable, while adding a contrastive loss to push positive and negative latent encodings apart. Executing a policy conditioned on an optimal latent trajectory then leads to preference-aligned behavior without a reward model. Since both the trajectory encoder and the policy use Transformer architectures, OPPO can also be seen as a generative PbRL approach.
>
> Beyond the PbRL setting, related ideas also appear in preference-based bandits or deterministic MDPs. For instance, DPO [4] directly optimizes generative models using preference pairs without reward supervision, and PFM [5] employs a conditional flow matching model to generate samples with higher preference scores. As discussed in Section A, these approaches operate under different settings and are therefore not reviewed as related work.

---

> ### Author Response · Authors · 2025-11-20
>
> ## Empirical Comparison with FTB
> Beyond the conceptual discussion, we also evaluated FTB [1] on the Meta-World medium-replay datasets (the datasets used in Section 5) with 1000 preference feedback. We used the official implementation of FTB and default hyperparameters, only adapting the episode length to be consistent with our datasets. The table below reports the results averaged over three random seeds.
>
> | Task | box-close | dial-turn | drawer-open | lever-pull | sweep-into | sweep | button-press-topdown | button-press-topdown-wall |
> | --- | --- | --- | --- | --- | --- | --- | --- | --- |
> | PVO | 58.72 $\pm$ 16.61 | 84.32 $\pm$ 4.71 | 100.00 $\pm$ 0.00 | 96.64 $\pm$ 2.84 | 27.84 $\pm$ 4.36 | 94.08 $\pm$ 6.31 | 43.36 $\pm$ 19.03 | 48.16 $\pm$ 14.54 |
> | FTB | 0.00 $\pm$ 0.00 | 0.00 $\pm$ 0.00 | 97.60 $\pm$ 3.39 | 0.00 $\pm$ 0.00 | 62.67 $\pm$ 23.71 | 0.00 $\pm$ 0.00 | 0.00 $\pm$ 0.00 | 0.00 $\pm$ 0.00 |
>
> Interestingly, FTB performs very well on *drawer-open* and even outperforms PVO on *sweep-into*, but collapses to 0% success on all other tasks. We note that a 0% success rate does not necessarily mean that the method failed to learn at all – we do observe nontrivial improvements in episodic returns during training – but rather that the learned policies rarely satisfy the success criteria on these tasks.
>
> Overall, these results suggest that **FTB can be effective on certain tasks, yet exhibit substantial sensitivity to the dynamics and data distribution**. One possible explanation is that FTB ultimately relies on imitation of generated trajectories: when the dataset contains sufficiently good trajectories, the filtering-and-cloning procedure may succeed, whereas in tasks where high-quality demonstrations are sparse, the model may end up cloning suboptimal behaviors. In contrast, **PVO achieves consistently strong performance across all tasks**, including those where FTB collapses, which aligns with our main goal: providing a simple, value-based offline PbRL algorithm that is both theoretically grounded and practically robust.
>
> We thank the reviewer for prompting this broader empirical comparison. The results with FTB have been added in Appendix F.2.
>
>
> [1] Zhilong Zhang, Yihao Sun, Junyin Ye, Tian-Shuo Liu, Jiaji Zhang, and Yang Yu. Flow to better: Offline preference-based reinforcement learning via preferred trajectory generation. In The Twelfth International Conference on Learning Representations, 2024.
>
> [2] Yachen Kang, Diyuan Shi, Jinxin Liu, Li He, and Donglin Wang. Beyond reward: Offline preference-guided policy optimization. In International Conference on Machine Learning, pp.15753–15768. PMLR, 2023.
>
> [3] Furuta, H., Matsuo, Y., and Gu, S. S. Generalized decision transformer for offline hindsight information matching. arXiv preprint arXiv:2111.10364, 2021.
>
> [4] Rafael Rafailov, Archit Sharma, Eric Mitchell, Christopher D Manning, Stefano Ermon, and Chelsea Finn. Direct preference optimization: Your language model is secretly a reward model. Advances in Neural Information Processing Systems, 36, 2024.
>
> [5] Minu Kim, Yongsik Lee, Sehyeok Kang, Jihwan Oh, Song Chong, and Se-Young Yun. Preference alignment with flow matching. Advances in Neural Information Processing Systems, 37:35140–35164, 2024.

---

### Official Review · Reviewer_23sR · 2025-10-30

**Soundness:** 4
**Presentation:** 3
**Contribution:** 3
**Rating:** 6
**Confidence:** 4

**Summary:**

This paper studies offline preference-based reinforcement learning and introduces Preference-based Value Optimization (PVO). The method directly aligns the learned value function with the reward inferred from human preferences through a novel value alignment loss, ensuring consistency between value estimation and preference supervision. The authors provide theoretical guarantees and strong empirical results, showing that PVO achieves stable and competitive performance across continuous-control benchmarks.

**Strengths:**

- The proposed value alignment loss is conceptually sound, novel, and powerful enough to achieve strong results.

- The empirical evaluation is comprehensive and consistent, demonstrating that PVO outperforms existing preference-based approaches across multiple continuous-control benchmarks.

- The paper is clearly written and well-organized, offering valuable intuition on how human preference signals can be effectively integrated into value-based offline reinforcement learning.

**Weaknesses:**

- The reward learning module closely follows standard preference-based MLE approaches and thus contributes limited novelty in this part.

**Questions:**

- Line 3 of Algorithm 1 differs from Eq. (4); there appears to be a sign inconsistency (the “+” and “−” symbols might be reversed).
- The proposed value alignment loss appears conceptually related to the value function inconsistency introduced in VIPO: Value-Inconsistency Penalized Offline Reinforcement Learning. A discussion clarifying the connection or distinction between these two ideas would be appreciated.

---

> ### Author Response · Authors · 2025-11-19
>
> Thank you for dedicating your time and expertise to reviewing our work and for offering thoughtful and encouraging feedback. Below, we respond to your comments in detail.
>
> ## On the Reward Model
> While PVO employs a standard MLE-based reward model, we would like to emphasize that this does not limit the novelty or contribution of our work.
> - **Core contribution.** The central contribution of PVO lies in the value alignment loss, which provides a **principled objective for learning values consistent with preference feedback**. As discussed in Section 3.1, this formulation is motivated by the fact that preference-based RL satisfies a distinct reward concentration bound (Lemma D.4), making the standard TD loss unsuitable. By combining the concepts of induced reward (Definition 1) and the reward concentration, our value alignment loss enables preference-consistent value learning, leading to both provable sample complexity guarantees and strong empirical performance.
> - **Orthogonality to reward learning.** In practice, the value alignment loss can be defined for any given reward model. Hence, our method is **orthogonal to the specific choice of reward learning algorithm**. One can readily integrate alternative preference-learning techniques [4,5,6,7] in place of MLE without affecting the formulation of PVO.
> - **Rationale for using MLE.** From a theoretical perspective, MLE-based reward modeling is the standard approach in preference-based RL and is commonly adopted in algorithms that provide theoretical guarantees [1,2,3]. We adopt MLE primarily to establish the concentration bound (Lemma D.4), rather than as a source of algorithmic novelty.
>
>
> ## Q1. Typo in Line 3 of Algorithm 1
> We sincerely appreciate your careful reading and for bringing this to our attention. There was indeed a typo in Line 3 of Algorithm 1. As you pointed out, the sign must be reversed to be consistent with Equation (4). We fixed this typo in the revised version.
>
> ## Q2. Comparison with VIPO
> We thank the reviewer for raising interesting connections to VIPO [8]. Below we clarify the similarities and key differences between VIPO and our proposed PVO.
>
> **Commonalities.**
> Both VIPO and PVO aim to learn a reliable value function and policy from offline datasets, addressing the fundamental challenge of distribution shift. Each incorporates mechanisms to mitigate this issue—VIPO through model uncertainty penalization, and PVO through expectile regression. Also, both frameworks involve mutual dependencies between model and value learning: VIPO reduces value inconsistency to improve model reliability, while PVO leverages the learned transition model to define the value alignment loss.
>
> **Differences.**
> The major difference lies in the source of feedback. VIPO assumes access to step-wise rewards as in standard RL, whereas PVO is trained solely from preference feedback. Consequently, as emphasized in Section 3, PVO replaces the step-wise TD loss with a value alignment loss derived from preference feedback.
> Second, VIPO is a model-based algorithm that utilizes model rollouts for policy improvement, while PVO does not rely on model rollouts. Our transition model approximates only one-step dynamics and, in the practical implementation (Section 3.4), the model expectation is replaced by samples from the dataset.
> Finally, VIPO penalizes value inconsistency to stabilize model learning, whereas PVO adopts standard maximum likelihood estimation for simplicity and theoretical tractability. This choice facilitates the theoretical analysis, but in practice PVO can incorporate more sophisticated modeling techniques—or even omit the model entirely, as in Section 3.4.
>
> We view these two approaches as complementary in addressing reliability in offline value learning, though under distinct supervision regimes (reward vs. preference). We hope the above clarification helps delineate the conceptual differences between VIPO and PVO.

---

> ### Author Response · Authors · 2025-11-19
>
> [1] Wenhao Zhan, Masatoshi Uehara, Nathan Kallus, Jason D. Lee, and Wen Sun. Provable offline preference-based reinforcement learning. In The Twelfth International Conference on Learning Representations, 2024.
>
> [2] Hyungkyu Kang and Min-hwan Oh. Adversarial policy optimization for offline preference-based reinforcement learning. In The Thirteenth International Conference on Learning Representations, 2025.
>
> [3] Alizée Pace, Bernhard Schölkopf, Gunnar Ratsch, and Giorgia Ramponi. Preference elicitation for offline reinforcement learning. In The Thirteenth International Conference on Learning Representations, 2025.
>
> [4] Jongjin Park, Younggyo Seo, Jinwoo Shin, Honglak Lee, Pieter Abbeel, and Kimin Lee. SURF: Semi-supervised reward learning with data augmentation for feedback-efficient preference-based reinforcement learning. In International Conference on Learning Representations, 2022.
>
> [5] Daniel Shin, Anca Dragan, and Daniel S. Brown. Benchmarks and algorithms for offline preference-based reward learning. Transactions on Machine Learning Research, 2023.
>
> [6] Minyoung Hwang, Gunmin Lee, Hogun Kee, Chan Woo Kim, Kyungjae Lee, and Songhwai Oh. Sequential preference ranking for efficient reinforcement learning from human feedback. Advances in Neural Information Processing Systems, 36, 2024.
>
> [7] Heewoong Choi, Sangwon Jung, Hongjoon Ahn, and Taesup Moon. Listwise reward estimation for offline preference-based reinforcement learning. In Forty-first International Conference on Machine Learning, 2024.
>
> [8] Xuyang Chen, Guojian Wang, Keyu Yan, and Lin Zhao. Vipo: Value function inconsistency penalized offline reinforcement learning. arXiv preprint arXiv:2504.11944, 2025.

---

> > ### Comment · Reviewer_23sR · 2025-11-26
> >
> > Thanks for your response. I have no other concerns and have decided to maintain my positive score.

---

### Meta-Review · Area_Chair_9oDr · 2026-01-04

**Summary:**

This submission studies offline preference-based RL and proposes Preference-based Value Optimization (PVO), which introduces a value alignment loss (built on an induced-reward formulation) to obtain a method that is both practically stable and theoretically grounded, with a claimed rate-optimal sample complexity and strong empirical performance across continuous-control benchmarks.
Across reviews, the key concerns are concentrated in three areas:
1. Novelty from prior work.
Reviewer frBc argues the algorithm is “essentially consistent with IQL” and that the theory is similar to prior provable offline PbRL work (Zhan et al.). Reviewer 23sR notes limited novelty in the reward learning module (MLE is standard), while still being positive about the value alignment contribution.
2. Presentation issues.
Reviewer 23sR notes a sign inconsistency in Algorithm 1 and requests clarification of the relation to VIPO. Reviewer BMtm points out formula-formatting issues.
3. Baseline completeness / fairness.
Reviewer BMtm wants comparison to generative PbRL methods (GAN/diffusion-style) that avoid an intermediate reward model. Reviewer frBc specifically requests a fairer IQL comparison (matching the advantage-weight parameter) and asks what role the transition model plays in practice.

Given that (i) all reviewer scores converge to marginally above threshold, (ii) the rebuttal resolves the concrete correctness/clarity issues and strengthens baselines, and (iii) the remaining concern is primarily about degree of novelty rather than soundness, I recommend Accept, with an emphasis in the final version on clearer positioning versus IQL and prior provable offline PbRL.

**Reviewer Concerns:**

Concerns addressed by the rebuttal:
1. Reviewer 23sR: sign error + VIPO connection. The authors fix the sign issue and provide a detailed differentiation from VIPO.
2. Reviewer BMtm: formatting. Authors correct the noted formatting inconsistency.
3. Reviewer BMtm: missing generative-method context/baselines. Authors expand discussion of generative PbRL and add an empirical comparison to FTB.
4. Reviewer frBc: motivation, IQL fairness, and transition-model role. Authors give an explanation tying the value alignment loss to PbRL concentration bounds, run additional IQL experiments with matched advantage-weight parameter, and clarify that the transition model is mainly for theory while a sample-based approximation suffices in practice.

Concerns not fully addressed
1. Residual novelty concerns relative to IQL / prior PbRL theory. While the rebuttal clarifies distinctions and adds experiments, reviewer frBc still conclude that related work already explored similar analysis directions and integration of transition models, and therefore view the incremental contribution as limited.

**Reviewer Scores:**

Reviewer 23sR (6 to likely 6). The reviewer explicitly state they have no further concerns and would maintain their positive score.

Reviewer BMtm (6 to likely 6, possibly 7). Given that their concerns are mostly (i) minor presentation fixes and (ii) adding generative-method discussion/baselines—both addressed with added discussion and an FTB experiment —my expectation is they would at least maintain the 6.

Reviewer frBc (4 to 6 already; likely remains 6). Reviewer frBc raise the score to 6, but still voice remaining novelty limitations; further discussion would most likely keep them at 6 rather than increase further.

---

### Decision · Program_Chairs · 2026-01-26

Accept (Poster)